# Towards Understanding Multi-Round Large Language Model Reasoning: Approximability, Learnability and Generalizability

## Abstract

Recent advancements in cognitive science and multi-round reasoning techniques for Large Language Models (LLMs) suggest that iterative thinking processes improve problem-solving performance in complex tasks. Inspired by this, approaches like Chain-of-Thought, debating, and self-refinement have been applied to auto-regressive LLMs, achieving significant successes in tasks such as mathematical reasoning, commonsense reasoning, and multi-hop question answering. Despite these successes, the theoretical basis for how multi-round reasoning enhances problem-solving abilities remains underexplored. In this work, we investigate the approximation, learnability, and generalization properties of multi-round auto-regressive models. We show that Transformers with finite context windows are universal approximators for steps of Turing-computable functions and can approximate any Turing-computable sequence-to-sequence function through multi-round reasoning. We extend PAC learning to sequence generation and demonstrate that multi-round generation is learnable even when the sequence length exceeds the model's context window. Finally, we examine how generalization error propagates across rounds, and show how the aforementioned approaches can help constrain this error, ensuring outputs stay within an expectation boundary. This work sheds light on the systemic theoretical foundations of multi-round sequence learning and reasoning, emphasizing its role in inference complexity.

## 1 Introduction

Cognitive science suggests that humans typically require multiple rounds of thinking to arrive at correct conclusions, especially when dealing with complex problems (Nelson, 1990). Inspired by this principle, recent multi-round reasoning techniques, such as Chain-of-Thought (Wei et al., 2022b), debating (Khan et al., 2024), and self-refinement (Madaan et al., 2023b; Liu et al., 2024a), applied to auto-regressive Large Language Models (LLMs), have achieved significant success across various reasoning tasks, including mathematical problem solving (Cobbe et al., 2021), commonsense reasoning (Talmor et al., 2019), scientific question answering (Clark et al., 2018), and multi-hop question answering (Yang et al., 2018). This exceptional capability is widely attributed to the in-context learning abilities of auto-regressive language models (Li et al., 2024a; Yang et al., 2024).

In-context learning ability has led to the conjecture that auto-regressive generative models can simulate Turing machines (Li et al., 2024b; Merrill & Sabharwal, 2024). Schuurmans (2023) has shown that large language models, particularly when augmented with memory, can execute complex computational processes that resemble the behavior of Turing machines. Malach (2024) has explored the theoretical underpinnings of auto-regressive models and their connection to universal computation, showing that even simple next-token predictors can, under the right conditions, emulate the behavior of Turing machines. Li et al. (2024b) illustrates the solvability of the problems that belong to $\text{AC}^0$, a proper subset of $\text{NC}^0$, via Chain-of-Thought. Yet, the existence of some strong assumptions of these works, such as the dependence on external memory (Schuurmans, 2023), the presence of Chain-of-Thought in the training data (Malach, 2024), or the infinite number of layers (Yun et al., 2020), do not explain well the actual working conditions of realistic multi-round language models.

To make matters worse, the computability perspective does not provide a plausible explanation for how machine learning models learn. This is because learning ability and reasoning ability are not inherent properties of a computable class. An extreme example is that even games like Magic: The Gathering and Super Mario can be Turing complete (Churchill et al., 2020). But we're not going to build a generalized AI out of a card game. Beyond the approximation of Turing machines, we need to understand (1) the ability of multi-round auto-regressive language models to approximate functions, (2) whether such an ability is learnable, and what its learning complexity is, and (3) the ability to generalize when inferring with imperfectly-trained language models in reality.

In this work, we systematically investigate why multi-round reasoning can improve the overall large language model problem-solving capability. Particularly, we study the approximation, learnability, and generalization abilities of auto-regressive generative models with a limited context window. Unlike previous theories, this series of studies corresponds closely to real-world scenarios, providing empirical guidance on training and inference auto-regressive generative language models.

We begin with the approximation ability of auto-regressive Transformers. For the approximation ability, we show that Transformers with finite context window size are universal approximators for some steps of a Turing-computable function. Further, we prove that any Turing-computable sequence-to-sequence function can be approximated by a multi-round auto-regressive generation process. This demonstrates the feasibility of the current dominant language models for solving problems. It is worth noting that although this finding does not directly explain the problem-solving ability of language models, it is a cornerstone of the generalizability of auto-regressive models.

Next, we turn our attention to learnability, which is a critical aspect of understanding how auto-regressive language models, particularly in a multi-round reasoning context, gain their problem-solving capabilities. We first expand probably approximately correct (PAC) learning (Valiant, 1984) to finite-size window next token prediction, and to auto-regressive sequence generation. Beyond that, we generalize the finite-window sequence learnability to the case of exceeding the window size by means of multi-round language generation. The results show that even if the required sequence length exceeds the maximum window of the auto-regressive language model, the model remains learnable for long sequence complex problems. The sample complexity required to learn the ability to auto-regressively predict an entire long sequence will increase dramatically compared to simply making a single-word prediction. Further, we show that training with the multi-round generation paradigm has an exponential impact on the sample complexity w.r.t the number of rounds $R$.

Then, we focus on the generalization ability through multi-round reasoning. In particular, we show that the generalization error of the model grows with the propagation rounds. Through our analysis, we can conclude that as the number of rounds of model generation increases, eventually the answers it obtains will diverge. Nevertheless, we can still constrain the state of the intermediate process, e.g., by providing some hints in the multi-round dialogues, to control the generalization of the model and induce it to the answer we want. We point out that prompting tricks like Chain-of-Thought, self-refinement, and multiple rounds of dialogue serve to constrain the generalization error during the inference process so that the answers generated are within our expectations.

**Contributions.** Overall, we make the following contributions:

- We comprehensively investigate the approximation, learning, and generalization capabilities of finite context auto-regressive language models from a theoretical perspective.
- We theoretically identify a dramatic increase in the sample complexity required to induce a finite window next-word prediction into sequence generation and remark that this increase can be mitigated by introducing multiple rounds of sub-sequence generation.
- We theoretically analyze the inter-round propagation mechanism of the generalization error during the multi-round generation process. We also point out that without intervention in the generation of multiple rounds, their cumulative error will not guarantee controllability.

The remaining paper is structured as follows: In Section 2, we discuss related research on multi-round language generation, language model learning, and generalization capabilities, followed by an introduction to some relevant lemmas and definitions in Section 3. We theoretically analyze the approximation capabilities of multi-round Transformers in Section 4, the learnability of language generation with auto-regressive language models in Section 5, and the error propagation of generalization in Section 6. We conclude and provide some insights and future directions in Section 7.

## 2 RELATED WORKS

### 2.1 MULTI-ROUND LANGUAGE MODEL GENERATION

It has been widely noted that with generative language models, it is possible to get the desired answer through multiple rounds of interaction. Based on this core idea, several types of multi-round generation methods have been proposed to solve a wide range of problems. These methods include:

**(1) Chain-of-Thought**. Wei et al. (2022b) proposes chain-of-thought (CoT), a prompt strategy that guides the LLM to generate regularized intermediate reasoning steps that lead from the initial question to the final answer. CoT has been proven to enable LLMs to solve complex problems including dynamic programming, whereas non-CoT prompts fail to do so (Feng et al., 2023). Empirically, variants of CoT have been proposed, including faithful CoT that tries to avoid LLMs lying in the reasoning stage (Lyu et al., 2023), multimodal CoT that enables CoT for vision contexts (Zhang et al., 2023), and tree-of-thought (ToT) that builds up a complex reasoning tree (Yao et al., 2023).

**(2) Self-correction and Self-refinement.** LLMs' self-correction and refinement abilities have received significant attention recently (Madaan et al., 2023b; Shinn et al., 2023; Kim et al., 2023). Through explicit multi-round reflection, self-correction iteratively improves the accuracy, reliability, and robustness of LLM outputs, especially in complex reasoning tasks where the CoT reasoning process might be flawed (Madaan et al., 2023b), by eliminating hallucinations (Liu et al., 2024b).

**(3) Multi-Agent Debating**. Apart from improving a single-agent LLM system, another line focuses on the collaboration of multiple LLM agents (Li et al., 2023; Wei et al., 2023; Hao et al., 2023). Findings suggest that multi-agent collaboration through both debating or even majority voting can often outperform single-agent LLM with explicit constraints (Huang et al.; Wu et al., 2023). Khan et al. (2024) further shows that debating with more persuasive LLMs results in better answers.

Overall, the essence of these methods is to impose internal or external interventions on multi-round sequence generation, which is an optimization constraint in the generalization process.

### 2.2 THEORETICAL ANALYSIS ON LANGUAGE MODELS

**Approximation Ability of Transformers.** The approximation capabilities of Transformer architectures have been extensively studied in recent years. Yun et al. (2020) establishes that Transformers are universal approximations of continuous permutation equivariant sequence-to-sequence functions. Wei et al. (2022a) introduces a statistically meaningful approximation, demonstrating that overparameterized neural networks can approximate Boolean circuits and Turing machines with generalization guarantees. Schuurmans (2023) proves that Transformers can simulate Turing machines in the presence of conditional memory. In terms of limitations, Dong et al. (2021) demonstrates that pure attention mechanisms could suffer from rank collapse, leading to a loss of expressive power with increased network depth. Hahn (2020) analyzes the theoretical limitations of self-attention in neural sequence models, highlighting challenges in modeling hierarchical structures. Cai (2024) proves that the composition of words in a finite vocabulary can approximate any continuous function in a compact domain. Unlike their setting, we show that a finite context Transformer can approximate a Turing Machine by up to infinite rounds of generation.

**Simulating Turing Machine with Neural Networks.** Numerous scholars have proposed using neural networks to simulate Turing machines to verify the computational power of neural networks (Siegelmann & Sontag, 1992; Pérez et al., 2021; 2019; Wei et al., 2022a; Graves, 2014). Siegelmann & Sontag (1992) point out that recurrent neural networks (RNNs) of infinite precision can make simulations of Turing machines, implying the Turing-completeness of RNNs. Chung & Siegelmann (2021) further implements a finite precision RNN simulation of Turing Machine using an external memory module. However this Turing-complete property cannot be directly inherited to a finite-window Transformer, because the recurrent learning model in RNNs essentially aggregates information from time 0. While the Transformer model via self-attention does not pass on the information, but rather, it is in the form of KV cache (Ge et al., 2024), which results in information beyond the context window being encoded only on newly generated tokens, a completely different computational paradigm from RNNs. For the Turing completeness of Transformer, Bhattamishra et al. (2020) prove that finite-precision Transformers are **not** Turing-complete. Pérez et al. (2021) prove that only hard-attention can be Turing complete. But Transformers still can approximate the

simulation of Turing Machine within a certain margin of error (Wei et al., 2022a). Yet, these studies have not examined the relationship between approximation precision and quantization precision. We reveal in this work the approximation of a Turing machine that can be reached through multiple rounds of Transformer's generation, as well as a high-level relationship between the precision of Transformer's numerical quantization and the approximation error tolerance.

**Learnability and Sample Complexity.** Probably Approximately Correct learning is a framework in computational learning theory that provides a formal way to understand how well an algorithm can learn a target function from a set of examples. The concept was introduced by Valiant (1984) and helped analyze how quickly and accurately a learning algorithm can generalize from data. For sequence modeling, Schuurmans & Greiner (1995) proposes a kind of sequential PAC learning where online stopping rules are used to minimize the number of samples required for PAC learning. Gavaldà et al. (2006) studies the PAC learnability of hidden Markov models. Malach (2024) extends this framework to the prediction of the next token, but there are assumptions of infinite context-awareness, as well as ignoring error propagation. So far, to the best of our knowledge, there are no previous works that study the PAC learnability and sample complexity of long sequence generation with a context window-limited auto-regressive generative model. Making up for the shortcomings of the above works, in this paper, we study the PAC learnability of sequence generation and apply it to a multi-round generation task to theoretically qualitatively study the effect of multi-round generation on the sample complexity of the learning of the long sequence generation task.

## 3 PRELIMINARY

Let $\Sigma$ be a finite alphabet, and let $\Sigma^*$ denote the set of all finite sequences over $\Sigma$. Consider a distribution $\mathcal{D}$ over input-output sequence pairs $(x, y) \in \Sigma^* \times \Sigma^*$. Let $f : \Sigma^* \to \Sigma^*$ be a target sequence-to-sequence function belonging to a class $\mathcal{F}$.

**Limited Context Window.** An auto-regressive model with a limited context window of size $k$ generates an output sequence $y = (y_1, y_2, \ldots, y_n)$ token by token. At each time step $t$, the model predicts the next token $y_t$ based solely on a context window $c_t$ of at most $k$ tokens. This context window $c_t$ consists of a combination of: Up to the last $k$ tokens from the input sequence $x$, specifically $x_{\max(1,t-k)}^{t-1}$. and previously generated output tokens $y_1^{t-1}$, limited to the most recent $k$ tokens.

Formally, the context window at time $t$ is defined as:

$$c_t = \left( x_{\max(1,t-k)}^{t-1}, \; y_{\max(1,t-k)}^{t-1} \right),$$

where $x_a^b$ denotes the subsequence $(x_a, x_{a+1}, \ldots, x_b)$.

**Sequential PAC Learnability.** A class $\mathcal{F}$ is efficiently Sequential PAC learnable if there exists an efficient algorithm that finds with high probability a sequence generator with low error. We formally capture this definition as follows:

**Definition 3.1** (Sequential PAC learnability)**.** *A class $\mathcal{F}$ of sequence-to-sequence functions is PAC-learnable with an auto-regressive model of context window size $k$ if there exists a learning algorithm $A$ and a polynomial function $p(\cdot, \cdot, \cdot)$ such that for any $\epsilon > 0$, $\delta > 0$, and target function $f \in \mathcal{F}$, given a sample of at least $m \geq p(1/\epsilon, 1/\delta, k)$ i.i.d. examples $\{(x^{(i)}, y^{(i)})\}_{i=1}^m$ drawn from $\mathcal{D}$, the algorithm $A$, operating under the context window limitation, outputs a hypothesis $h$ such that with probability at least $1 - \delta$:*

$$\Pr_{(x,y) \sim \mathcal{D}} [d(h(x), y) \neq 0] \leq \epsilon,$$

*where $d$ is distance measure of discrepancy between predicted sequence $h(x)$ and true sequence $y$.*

**Rademacher Complexity** is a measure to quantify the capacity of a class of functions based on its ability to fit random noise, which is formally represented in the following Definition 3.2. It helps to analyze how well a model class can learn from the data.

**Definition 3.2.** *[Rademacher Complexity (Bartlett & Mendelson, 2002)] Let $\mathcal{F}$ be a class of real-valued functions on a domain $\mathcal{X}$, and let $\{x_1, x_2, ..., x_n\}$ be a set of $n$ independent and identically distributed (i.i.d.) samples drawn from a distribution over $\mathcal{X}$. The empirical Rademacher complexity*

of the class $\mathcal{F}$ with respect to the sample $\{x_1, x_2, ..., x_n\}$ is defined as:

$$\hat{\mathcal{R}}_n(\mathcal{F}) = \mathbb{E}_\sigma \left[ \sup_{f \in \mathcal{F}} \frac{1}{n} \sum_{i=1}^n \sigma_i f(x_i) \right]$$

where $\sigma_1, \sigma_2, ..., \sigma_n$ are independent Rademacher random variables, which take values in $\{-1, 1\}$ with equal probability $\frac{1}{2}$. And $\mathbb{E}_\sigma$ denotes the expectation over these random variables.

## 4 APPROXIMABILITY

### 4.1 TRANSFORMER CAN APPROXIMATE FINITE STEPS OF TURING-MACHINE

In this section, we present our theory showing that auto-regressive Transformers with a limited context window are universal approximators of any Turing computable functions. Although the simulation of a Turing machine does not guarantee that the model will always find the correct answer, it is a necessary condition for the language model to be able to effectively generate the expected text. By demonstrating that the Transformer's computational power is capable of approximating a Turing machine, we show that it reaches Turing Machine's limits when handling complex sequential tasks.

We begin by encoding the computation of a corresponding Turing machine $M$ that computes $f$. Each computation step of $M$ is represented as a configuration $C_t$, encapsulating the current state, tape contents, and head position. These configurations form a sequence $C_0, C_1, \ldots, C_T$, where $C_0$ is the initial configuration based on the input $x$, and $C_T$, where: $C_0 = \{x, q_0, \#\}$, $C_{t+1} = \delta(C_t)$, and $C_T = \{\cdot, q_{\text{accept}}, \cdot\}$ encodes the halting configuration producing $f(x)$, represents the halting configuration yielding $f(x)$, where $\#$ represents empty tape. The transition function $\delta$ of $M$ updates only a finite region of the tape based on the current state and symbol under the head: $\delta : \Gamma^* \mathbb{Q} \Gamma^* \to \Gamma^* \mathbb{Q} \Gamma^*$, where $\Gamma*$ is empirical tape symbol space, and $\mathbb{Q}$ is TM state space

We first simulate the finite steps of a Turing Machine. The following lemmas show that several steps of the Turing machine can be simulated by an auto-regressive generative Transformer.

**Lemma 4.1.** *Let $\mathcal{M}$ be any deterministic Turing Machine that operates in $S$ steps. For any $\epsilon > 0$, there exists a Transformer model $\mathcal{T}$ characterized by a finite number of layers $L$, layer dimension $d$, attention window size $k$, and quantization levels $Q$, such that for all computational steps $s \leq S$, the state of $\mathcal{T}$ approximates the state of $\mathcal{M}$ at step $s$ within an error bound $\epsilon$.*

$$\forall \epsilon > 0, \ \exists \mathcal{T} \text{ with parameters } (L, d, k, Q) \text{ satisfying } \begin{cases} d \geq \log_2(|\mathbb{Q}|) + k \cdot \log_2(|\Gamma|), \\ Q \geq e^{\frac{C''' \cdot L \cdot d \cdot k}{\varepsilon}} \end{cases}$$

*such that $\forall s \leq S, \ d(H_s, \phi(C_s)) \leq \epsilon$.*

A key feature of the Turing Machine is its ability to store and access a large amount of information via its "Tape". When Transformers simulate Turing Machines through multi-round generation, they rely on the attention mechanism to store and retrieve information from previous generations. This suggests that Transformers can function as dynamic memory systems during the multi-round process, akin to the way a Turing Machine reads and writes on its tape.

**Lemma 4.2.** *The maximum number of computational steps $S_{max}$ that $\mathcal{T}$ can approximate while maintaining this error bound scales asymptotically as*

$$S_{max} \in \Theta \left( L \cdot d \cdot k \cdot \log(Q) \right).$$

Lemma 4.2 illustrates that a finite-precision, finite-depth, finite-width Transformer has only limited problem-solving capabilities. We demonstrate Lemma 4.1 and 4.2 in Appendix A and B.

### 4.2 MULTI-ROUND TRANSFORMERS ARE TURING-MACHINE APPROXIMATOR

We now consider the more far-reaching case: even if the Turing machine does not reach the halting condition within the Transformer's maximum generation window, we show that the Transformer can still approximate the simulation of the Turing machine through multiple rounds of generation.

**Theorem 4.3** (Approximability). *For any Turing-computable sequence-to-sequence function $f$ : $\Sigma^* \to \Sigma^*$, and for any error tolerance $\epsilon > 0$, there exists a multi-round sequence-to-sequence process utilizing an auto-regressive Transformer with a limited context window of size $k$ that approximates $f$ within error $\epsilon$.*

Theorem 4.3 can be inferred by induction directly from Lemma 4.1. The core idea for the proof is to consider that a Turing machine will halt within a sequence of finite length $T$, by simulating the Turing Machine within $R = \lceil T/s \rceil$ rounds and using induction for error propagation, the approximation error can be controlled within the tolerance. See detailed proof in Appendix C.

Thus, we conclude that any Turing-computable sequence-to-sequence function $f : \Sigma^* \to \Sigma^*$ can be universally approximated by a multi-round sequence-to-sequence refinement process utilizing an auto-regressive Transformer with a limited context window of size $k$, achieving the desired approximation within error $\epsilon$. Through multi-round generation, the Transformer moves beyond static one-shot input-output mappings and instead continuously adjusts its generation, similar to how human reasoning progresses. This dynamic computational ability is crucial for cognitive tasks as it allows the model to update its internal state and strategy during the process. This means that Transformers, beyond being powerful sequence generation models (e.g., for language translation), could potentially be applied to complex tasks such as cognitive reasoning, planning, and problem-solving.

## 5 LEARNABILITY

In Section 4.1, we illustrated the ability of the autoregressive Transformer to approximate a sequence just like a Turing machine, which is a preliminary indication of the inference potential of language models. However, we still do not know what scale of training the model needs to undergo to obtain such a capability. In this section, we explore the learning ability of autoregressive models for sequences of arbitrary length.

To get this point, we aim to demonstrate that auto-regressive sequence models, which possess universal approximation capabilities, can be sequential PAC-learnable under certain constraints. In order to ensure that a sequential model learns effectively from finite data, we need to determine the minimum sample size $m$ required to guarantee that the model's error on unseen data does not exceed a specified threshold $\epsilon$ with a high level of confidence $1 - \delta$. This involves deriving a bound on the generalization error using tools from statistical learning theory, such as Rademacher complexity (Bartlett & Mendelson, 2002) and spectral norm constraints(Bartlett et al., 2017).

### 5.1 BASIC ASSUMPTIONS AND LEMMAS

We consider an auto-regressive next-token prediction model with a context window size $k$. At each time step $t$, the model predicts the next element based on the previous $k$ elements in the sequence. The hypothesis class $\mathcal{H}_k$ now consists of functions $h : \Sigma^k \to \Sigma$, where $\Sigma^k \subset \Sigma^*$ is context window of the auto-regressive language model, belonging to the input space. We make the following assumption:

**Assumption 5.1** (Bounded Input Norms). *Each element $x_i$ in the sequence satisfies $\|x_i\| \leq R_x$, therefore each input sequence in the context window $x \in \Sigma^k$ has bounded norm $\|x\| \leq R_x\sqrt{k}$.*

This assumption is reasonable because, within a finite dictionary, we can always find a sufficiently large number, denoted as $R_x$, such that the norm of a finite-dimensional embedding is less than $R_x$.

**Assumption 5.2** (Lipschitz-Continuous Activation Functions). *The activation functions $\phi$ used in the neural network are $L_\phi$-Lipschitz continuous.*

This assumption is reasonable because commonly used activation functions such as GELU, ReLU, etc, conform to Lipschitz continuity.

**Assumption 5.3** (Lipschitz-Continuous Loss Function). *The loss function $\ell$ is L-Lipschitz with respect to its first argument and bounded by $C > 0$.*

This assumption is reasonable because commonly used loss functions such as cross-entropy conform to Lipschitz continuity and are bounded in practice, we will show this in Appendix L.

**Assumption 5.4** (Spectral norms (Bartlett et al., 2017)). *For layer $1 < l < l_{max}$, the spectral norms of the weight matrices $W_l$ in the neural network are bounded by a layer Boundary $B_l$ such that $\|W_l\|_2 \leq B_l$, and $B_{spec} = \prod_{l=1}^{l_{max}} B_l$.*

We first consider the learnability of the next token prediction. Within a finite context window k, consider the generalization bound defined by Rademacher complexity (Bartlett & Mendelson, 2002):

**Lemma 5.5** (Rademacher complexity boundary for next token prediction). *The Rademacher complexity of the hypothesis class $\mathcal{H}_k$ satisfies:*

$$\mathcal{R}_m(\mathcal{H}_k) \leq \frac{B_{spec} L_\phi^{l_{max}-1} R_x \sqrt{k}}{\sqrt{m}}$$

We provide the computation of Rademacher complexity for next-token prediction in Appendix D.

**Lemma 5.6.** *For the standard generalization bound via Rademacher complexity, we have the loss $L(h)$ with probability at least $1 - \delta$:*

$$L(h) \leq \hat{L}_S(h) + 2\mathcal{R}_m(\mathcal{H}_k) + C\sqrt{\frac{\log(1/\delta)}{2m}},$$

## 5.2 SAMPLE COMPLEXITY OF NEXT-TOKEN PREDICTION

Now, we show that in order to obtain the ability to predict the next token, an auto-regressive model should be trained with at least a certain sample complexity. For the generation of individual tokens, we do not consider error propagation for now. This paradigm is consistent with decoder-only autoregressive model training since each token is determined by up to k previous tokens.

**Theorem 5.7** (Sample Complexity for Next-token Learning). *To ensure that the expected loss $L(h)$ does not exceed $\epsilon$ with confidence at least $1 - \delta$, under perfect empirical risk minimization , the required sample size $m$ must satisfy:*

$$m \geq \frac{1}{\epsilon^2}\left[4L^2 B_{spec}^2 L_\phi^{2(l_{max}-1)} R_x^2 k + 4L B_{spec} L_\phi^{l_{max}-1} R_x C\sqrt{k}\sqrt{\frac{\log(1/\delta)}{2}} + \frac{C^2 \log(1/\delta)}{2}\right].$$

The proof of Theorem 5.7 can be done by solving inequality properties $L(h) \leq \epsilon$. We provide detailed proof in Appendix E. Specifically, the three terms are Capacity Term, Mixed Term, and Confidence Term, respectively. The Capacity Term is the dominant term for the sample complexity needed to reach the window size $k$, while the Confidence Term denotes the complexity needed to reach higher learning confidence $1 - \delta$, and the Mixed Term is a lower-order mixture of these two terms that does not dominate. The Capacity Term is the dominant term when we consider larger context window $k$ and moderate confidence level $\delta$. For simplicity, we combine the Mixed Term and the Confidence Term into one low-order term. Therefore, for single next token prediction, we have sample complexity as:

$$m \geq \frac{1}{\epsilon^2}\left[4L^2 B_{spec}^2 L_\phi^{2(l_{max}-1)} R_x^2 k + \text{low order term}\right].$$

## 5.3 SAMPLE COMPLEXITY OF SEQUENCE GENERATION

Next, we consider the generation of a sequence of arbitrary length $T$. When the model generates sequences over $T$ time steps, the cumulative error over the sequence is of interest. Due to the dependencies introduced by using the model's own predictions as inputs in subsequent time, errors can compound over time. This phenomenon is known as error propagation in auto-regressive models (Wu et al., 2018). We bound the cumulative error carefully by considering the worst-case scenario where errors add up linearly so that cumulative error $\epsilon < \sum_{t=0}^{T} \epsilon_t$. For hypothesis $\mathcal{H}_k$, we recursively define the output at time $t$ to be $h^{(t)}(x_t) = h(x_t, (h^{\max(t-k,1)}(x_{\max(t-k,1)}), \cdots, h^{(t-1)}(x_{t-1}))$, starting with $h^{(1)}(x) = h(x)$. By extending Therorem 5.7 to sequence generation, we have the following:

**Theorem 5.8** (Sample Complexity for Sequence Learning). *For any sequence of length T, to ensure that the expected loss $L(h^{(T)})$ does not exceed $\epsilon$ with confidence at least $1 - \delta$, the required sample size $m$ must satisfy:*

$$m \geq \frac{\left(B_{spec}L_\phi^{l_{max}-1}\right)^{2T}}{\epsilon^2 \left(B_{spec}L_\phi^{l_{max}-1} - 1\right)^4} \left[4L^2 B_{spec}^2 L_\phi^{2(l_{max}-1)} R_x^2 k + low\ order\ term\right].$$

Theorem 5.8 tells us that the sample complexity of learning a sequence is far higher than the prediction of a single token. This complexity grows exponentially with the length of the sequence. The proof of Theorem 5.8 can be found in Appendix F. We next consider that if the learning of a sequence of length $T$ is performed in $R$ rounds, where each round involves generating a sequence of length $T/R$, its sample complexity is affected by $R$. [1]

**Theorem 5.9** (Sample Complexity for Multi-Round Sequence Learning). *For any sequence of length T, if the sequence is dismantled to the R rounds learning, to ensure that the expected loss $L(h^{(T)})$ does not exceed $\epsilon$ with confidence at least $1 - \delta$, the required sample size $m$ must satisfy:*

$$m \geq \frac{\left(B_{spec}L_\phi^{l_{max}-1}\right)^{\frac{2T}{R}+2} \cdot R^2}{\epsilon^2 (B_{spec}L_\phi^{l_{max}-1} - 1)^4} \left[4L^2 B_{spec}^2 L_\phi^{2(l_{max}-1)} R_x^2 k + low\ order\ term\right].$$

The essence of Theorem 5.9 is that decomposing a large sequence learning problem into multiple rounds can significantly reduce the sample complexity required for effective learning. In multi-round training, the model learns $R$ smaller sequences of length $T/R$ per round, effectively distributing the task across rounds. This reduction in the learning burden for each round minimizes the potential for error propagation and avoids the exponential growth of sample complexity typically associated with longer sequences, as seen in Theorem 5.8. By breaking down the sequence, the single-round complexity decreases exponentially with $R$, while the cumulative complexity grows polynomially. This balance leads to an overall more efficient learning process, ensuring that the required sample size $m$ for a given confidence $1 - \delta$ and error threshold $\epsilon$ becomes more manageable. The proof of Theorem 5.9 is provided in Appendix G. This insight opens a path toward reducing the complexity of sampling, thereby optimizing training regimes in auto-regressive sequence models.

# 6 GENERALIZATION ABILITY

In this section, we discuss the propagation of the generalization error between each round of a multi-round sequence generative model. The generalization ability of an auto-regressive generative language model determines its ability to solve real problems by continuous generation for practical reasoning tasks. We focus on the inter-round propagation of errors when generating a long sequence via $R$-rounds, and how the accumulation of these errors acts.

## 6.1 THE PROPAGATION AND CUMULATION OF ERROR

We first consider the case where our model is generated in the $r$-th round with an aggregate error due to the dependence on the previously generated contexts. In this process, we consider the effect that the error of the previous round has on the current round.

**Lemma 6.1** (Aggregate Error). *For an auto-regressive generative model over R rounds, the generalization bound of the aggregate error for each round $r$ satisfies:*

$$L_r(h_r) \leq \sum_{i=1}^{r} \left(\left(\prod_{j=i+1}^{r} \gamma_j\right)\left(\hat{L}_{m,i}(h_i) + \epsilon_i\right)\right),$$

---

[1] It is important to note that, the difference between generating T tokens via R rounds and generating T tokens in 1 round is that the 1-round generation looks for T-independent tokens directly in the dictionary to complete the composition of the sequence, whereas the R round generation generates shorter sequences, and the learning of such shorter sequences requires a smaller sample complexity (by Theorem 5.8), but a certain amount of sample complexity to complete the reassembling of the shorter sequences. The idea of Divide and Conquer Cormen et al. (1994) is used here, although not exactly the same.

where $L_r(h_r) = \mathbb{E}_{(x^{(r)}, y^{(r)}) \sim \mathcal{D}^{(r)}}[\ell(h_r(x^{(r)}), y^{(r)})]$ is the loss of the model's output sequence in the $r$-th round from the expected output. $\hat{L}_{m,i}(h_i)$ is the empirical loss computed on $m$ samples. The $\gamma_r$ quantifies the impact of errors from round $r-1$ on round $r$. See Appendix H for proof.

**Theorem 6.2** (Cumulative error). *For an auto-regressive generative model over $R$ rounds, the generalization bound of the cumulative error satisfies:*

$$L(h_R) = \sum_{r=1}^{R} \lambda_r L_r(h_r) \leq \sum_{i=1}^{R} \Lambda_i \left( \hat{L}_{m,i}(h_i) + \epsilon_i \right).$$

where $\Lambda_i = \sum_{r=i}^{R} G_{r,i}$ represents the total influence that the generalization error at round $i$ has on the cumulative error across all subsequent rounds $r \geq i$, $\lambda_r$ are non-negative cumulative weights with $\sum \lambda_r = 1$. And here $G_{r,i} = \lambda_r \prod_{j=i+1}^{r} \gamma_j$ captures the influence of the generalization error at round $i$ on the loss at round $r$. Specifically, $G_{r,i}$ accounts for the cumulative error for how aggregate errors propagate from round $i$ through subsequent rounds up to round $r$. See Appendix I for proof.

## 6.2 CUMULATIVE ERROR INCREASES AS ROUNDS GROWS

We need to note that, in the real world, models often do not guarantee zero empirical loss ($\hat{L}_{m,i}(h_i) > 0$) due to limitations of existing optimization methods, model architecture, etc. Let's first assume that the error impact factor $\gamma = \gamma_r$ is uniform between each round, also a uniform influence factor $\lambda_r = \lambda, \forall r \in \{1, \cdots, R\}$ and a uniform lower bound $\eta \geq \hat{L}_{m,i}(h_i) + \epsilon_i$ for simplification. Denote the upper bound of cumulative error $L(h_R)$ as $\bar{L}(h_R) = \sum_{i=1}^{R} \Lambda_i \left( \hat{L}_{m,i}(h_i) + \epsilon_i \right)$. We now observe the trend of the cumulative error upper bound $\bar{L}(h_R)$ generated by $R$-rounds evolves as the number of rounds $R$ gradually tends to infinity.

**Theorem 6.3** (Divergence of Cumulative Error Upper Bound). *As the generation rounds $R$ increase to infinity, the upper bound of generation cumulative error $L(h_R)$ satisfies:*

$$\lim_{R \to \infty} \bar{L}(h_R) = \lim_{R \to \infty} \frac{\eta \lambda}{1 - \gamma} \sum_{i=1}^{R} \left( 1 - \gamma^{R-i+1} \right) \to \infty.$$

Theorem 6.3 implies that there is no supremum on the cumulative error as rounds $R$ increases. This means that, in reality, considering models with limited accuracy and limited training, we do not always get the results we expect using multiple rounds of inference. This may sound like a disappointing result, suggesting that if the model is allowed to keep generating round by round, its final generated content can be uncontrollable. But the good news is that, for most practical scenarios, the rounds are typically finite, hence the cumulative error is within an acceptable range. This suggests that even though the content the auto-regressive language model generates in a finite number of rounds may not be the precise solution to the problem that we expect, it is at least to a degree relevant to a specific topic that we expect. See Appendix J for proof of Theorem 6.3.

## 6.3 MULTI-ROUND GENERATION TECHNIQUES AS INTERVENTION

We now consider the use of techniques, such as Chain-of-Thought (Wei et al., 2022b) or self-correction (Madaan et al., 2023a), and multi-agent dialog (Khan et al., 2024) to intervene in the generation process of language models. Chain-of-thought restricts the intermediate steps in multiple rounds of generation by means of a prompt template of the solution idea, in order to block the effect of errors in this round on the subsequent generation. Self-correction limits the effect of errors by constantly making multiple small adjustments to the results obtained in the previous round. Multi-agent dialog, on the other hand, limits the aggregate error by the information given by other agents, thus reducing the error propagation. It is worth noting that these tricks are ultimately a kind of generative process intervention to reduce $\gamma_r$ to $\gamma'$ at certain rounds.

**Theorem 6.4.** *If we allow intervention several times during the generation process by making $\gamma_r$ decrease to $\gamma'$, then let $h_{i,r}$ be the number of hint rounds between $i+1$ and $r$. It can be shown that the reduction in cumulative error can be given by:*

$$\Delta L(h_R) = \sum_{i=1}^{R} (1 - \kappa_i) \Lambda_i \left( \hat{L}_{m,i}(h_i) + \epsilon_i \right)$$

*where $\kappa_i = \mathbb{E}_{r \sim \mu_i} \left[ \left( \frac{\gamma'}{\gamma} \right)^{h_{i,r}} \right]$, $\mu_i$ is a probability distribution over $r$ defined by: $\mu_i(r) = \frac{\lambda_r \gamma^{r-i}}{\Lambda_i}$.*

Theorem 6.4 means that if we obtain a good intervention such that the error propagation at certain rounds is effectively controlled by a smaller amount to $\gamma'$, the cumulative error of sequence generation will be effectively controlled. To achieve effective control over the cumulative error, i.e. large cumulative error reduction $\Delta L(h_R)$, we expect a smaller $\kappa_i$. This can be accomplished in two ways: (1) Improving the quality of hints: A good hint can effectively block error propagation in a single-round generation, leading to a smaller $\gamma'$, eventually smaller $\kappa_i$. (2) Increase the number of hints: By increasing the number of hints between round $i$ to $r$, we will get a larger $h_{i,r}$, which ultimately leads to a decrease in $\kappa_i$ in average regarding $i$. See Appendix K for proof of Theorem 6.4.

## 7 CONCLUSION

In conclusion, this work provides a comprehensive theoretical foundation for understanding the capabilities of multi-round reasoning in auto-regressive large language models. We have systematically explored the approximation, learnability, and generalization properties of these models, highlighting their potential to solve complex problems through iterative reasoning.

**Findings.** We demonstrate that Transformers with a finite context window can serve as universal approximators for Turing-computable functions, offering insights into their robust problem-solving capabilities in real-world tasks. Additionally, we have extended the PAC learning framework to account for sequence generation tasks, revealing the complexities involved in learning long sequences when context exceeds the model's window size. Moreover, our analysis of the generalization error in multi-round generation reveals that, without proper interventions, error accumulation could lead to divergence in the model's outputs. Techniques like Chain-of-Thought offer viable strategies to mitigate this, ensuring that the generated sequences remain within expected bounds.

**Practical Insights.** Our contributions not only advance our theoretical understanding of auto-regressive generative language models but also provide practical insights into improving model performance through multi-round reasoning interventions. For the model training stage, in order to reduce the sample complexity of training on long sequences, one can consider providing some decomposition methods for very long and complex task sequences during language model training, so that the long sequences are decomposed into multiple rounds of training on short sequences. In the inference process of the model, when we design a method that makes the model perform multi-round thinking, we should give more consideration to how to interrupt the propagation of cumulative errors to make the generated content more in line with our expectations.

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

## A PROOF OF LEMMA 4.1

### A.1 PROOF SKETCH

To prove the Lemma 4.1, we will:

1. Define an Encoding Function $\phi$ that maps TM configurations $C_t$ to Transformer hidden states $H_t$.

2. Demonstrate that each Transformer layer $\mathcal{T}^{(i)}$ can simulate one TM step.

3. Ensure the accurate encoding function $\phi$.

### A.2 ENCODING FUNCTION $\phi$

For all computational steps $s \in \{0, 1, 2, \ldots, S\}$, the Transformer's hidden state $H_s$ approximates the TM's configuration $C_s$ within the error bound $\epsilon$. Formally,

$$\forall s \in \{0, 1, 2, \ldots, S\}, \quad d(H_s, \phi(C_s)) \leq \epsilon$$

where $\phi : \{C_s\} \to \mathbb{R}^d$ is an encoding function that maps the TM's configuration to the Transformer's hidden state space. Pérez et al. (2021)'s conclusion ensures the existence of such a mapping $\phi$.

**Definition A.1.** *The encoding function $\phi$ of TM configurations is defined as:*

$$\phi(C_s) = [s_t \oplus \mathbf{0}_{d_s}] \oplus S_t,$$

*where $s_t$ is a one-hot representation of the current state, $\mathbf{0}_{d_s}$ is a zero vector to match the dimensions, and $S_t$ is the one-hot representation of tape symbols for up to k contexts with the current head position and their positional representation.*

This function ensures:

- Injectivity: $\phi(C_t) \neq \phi(C_{t'})$ for $C_t \neq C_{t'}$.
- Surjectivity: Every $H_t$ corresponds to some $C_t$.

To demonstrate Definition A.1, we first clarify the TM configuration. Formally, a TM configuration $C_t$ at time $t$ consists of:

- Current State $q_t$: We define The state of the TM at time $t$ as $q_t \in \mathbb{Q}$, where $\mathbb{Q}$ is a finite state set.
- Tape Contents $\Gamma_t$: A mapping from tape positions at time $t$ to symbols in $\Gamma$, where $\Gamma$ is a finite tape alphabet.
- Head Position $h_t$: The position of the tape head at time $t$.

In order to construct the encoding function, we need to consider TM's state embedding, tape's content embedding and position embedding.

(1) To get the state embedding, we assign a unique one-hot vector $e(q) \in \mathbb{R}^{|\mathbb{Q}|}$ to each state $q_t \in \mathbb{Q}$:

$$e(q_t) = [0, \ldots, 1, \ldots, 0]^T,$$

where the 1 is at the position corresponding to state $q_t$.

(2) To get the tape content's embedding, we first assign a unique one-hot vector $e(\gamma) \in \mathbb{R}^{|\Gamma|}$ to each tape symbol $\gamma \in \Gamma$:

$$e(\gamma) = [0, \ldots, 1, \ldots, 0]^T,$$

with the 1 at the position corresponding to symbol $\gamma$. Then we consider an acceptance window of size $k$, (which is the Transformer's max number of context tokens that can be dealt with simultaneously) centered at the head position $h_t$:

$$T_t = \left[ e(\gamma_{h_t - \lfloor k/2 \rfloor}), \ldots, e(\gamma_{h_t}), \ldots, e(\gamma_{h_t + \lfloor k/2 \rfloor}) \right] \in \mathbb{R}^{k \times |\Gamma|}.$$

This window captures the tape symbols around the head position.

(3) Finally we include a global positional encoding $p(i) \in \mathbb{R}^{d_p}$ for each relative position $i$ within the window:

$$P_t = [p(-\lfloor k/2 \rfloor), \ldots, p(0), \ldots, p(\lfloor k/2 \rfloor)] \in \mathbb{R}^{k \times d_p}.$$

The positional encoding helps the model distinguish positions within the acceptance window.

For each position, we concatenate the tape symbol embedding and its positional encoding:

$$S_t = \left[ e(\gamma_{h_t - \lfloor k/2 \rfloor}) \oplus p(-\lfloor k/2 \rfloor), \ldots, e(\gamma_{h_t}) \oplus p(0), \ldots, e(\gamma_{h_t + \lfloor k/2 \rfloor}) \oplus p(\lfloor k/2 \rfloor) \right] \in \mathbb{R}^{k \times (|\Gamma| + d_p)},$$

where $\oplus$ denotes concatenation.

Similarly, we create the encoding for the current state $q_t$:

$$s_t = e(q_t) \in \mathbb{R}^{|Q|}.$$

Finally, the full encoding $\phi(C_t)$ is obtained by forming a sequence of length $n = k + 1$ (state plus tape symbols):

$$\phi(C_t) = [s_t \oplus \mathbf{0}_{d_s}] \oplus S_t,$$

where $\mathbf{0}_{d_s}$ is a zero vector to match the dimensions.

For easier understanding of the following parts, we denote the sequence as:

$$\phi(C_t) = [x_0, x_1, \ldots, x_k],$$

where $x_0 = s_t \oplus \mathbf{0}_{d_s} \in \mathbb{R}^d$, and $x_i = e(\gamma_{h_t - \lfloor k/2 \rfloor + i - 1}) \oplus p(-\lfloor k/2 \rfloor + i - 1) \in \mathbb{R}^d$ for $i = 1, \ldots, k$.

For simplicity, assume $d$ is sufficiently large (we will define later in Appendix A.3.1) to accommodate all concatenated vectors without loss. The $d$ is also used to construct the hidden dimension of the Transformer, ensuring that embeddings and positional encoding fit within the hidden state.

### A.2.1 TRANSFORMER LAYER AS TM STEP SIMULATOR

In this subsection, we explore a correct but not necessarily optimal simulation scheme for the TM step. Each Transformer layer $\mathcal{T}^{(i)}$ performs the following operations to simulate one TM step:

A standard Transformer layer consists of:

- Multi-Head Self-Attention (MHSA) which computes attention over the input sequence.
- Feed-Forward Network (FFN) which applies a non-linear transformation to the outputs of the MHSA.
- Add & Norm which are residual connections and layer normalizations.

Our plan is to design the MHSA and FFN to simulate the TM's transition function.

### A.2.2 SELF-ATTENTION AND FFN COMPUTATION FLOW

Let's start with a quick introduction to Self-Attention. The self-attention mechanism computes attention scores between elements of the input sequence. Given an input sequence of vectors $X = [x_0, x_1, \ldots, x_k]$ whcih is consistent with $\phi(C_t)$, the self-attention computes:

$$Q = XW_Q, \quad K = XW_K, \quad V = XW_V,$$

where $W_Q, W_K, W_V \in \mathbb{R}^{d \times d}$ are projection matrices. Then the attention score is given by:

$$\alpha_{i,j} = \frac{q_i \cdot k_j}{\sqrt{d}} + M_{i,j},$$

where $q_i \in Q, k_j \in K$, and $M_{i,j}$ is the attention mask, and $\alpha_{i,j}$ is the unnormalized attention score. Then:

$$a_{i,j} = \frac{\exp(\alpha_{i,j})}{\sum_{l=0}^{k} \exp(\alpha_{i,l})},$$

where $a_{i,j}$ is the normalized attention weight. The final Attention output is given by:

$$\text{Attention}(X)_i = \sum_{j=0}^{k} a_{i,j} v_j.$$

The FFN is typically defined as:

$$\text{FFN}(u) = W_2 \cdot \sigma(W_1 u + b_1) + b_2,$$

where: $W_1 \in \mathbb{R}^{h \times d}$ and $W_2 \in \mathbb{R}^{d \times h}$ are weight matrices, $b_1 \in \mathbb{R}^h$ and $b_2 \in \mathbb{R}^d$ are biases, and $\sigma$ is an activation function (e.g., ReLU).

### A.2.3 CONSTRUCTION OF TRANSFORMER LAYER TO SIMULATE TM STEP

For TM simulation, we need to design the attention mechanism so that the state embedding $x_0$ attends to the tape symbol at the head position $x_{i_h}$, where $i_h = \lfloor k/2 \rfloor + 1$. And the tape symbol at the head position $x_{i_h}$ attends to the state embedding $x_0$. While the tape symbols not at the head position attend only to themselves.

**Lemma A.2.** *A Self-Attention layer is able to exchange numerical value of position $0$ and $i_h$, where $i_h = \lfloor k/2 \rfloor + 1$.*

*Proof.* For this propose, we then make the following construction:

First, we define the attention mask $M \in \mathbb{R}^{(k+1) \times (k+1)}$ as:

$$M_{i,j} = \begin{cases} 0, & \text{if } (i = 0 \text{ and } j = i_h) \text{ or } (i = i_h \text{ and } j = 0) \text{ or } (i = j), \\ -\infty, & \text{otherwise.} \end{cases}$$

Then we assume $W_Q = W_K = W_V = I$ (identity matrix), and biases $b_Q = b_K = b_V = 0$. Therefore:

$$q_i = x_i, \quad k_j = x_j, \quad v_j = x_j.$$

Following the computation flow of Self-Attention, the attention scores are:

$$\alpha_{i,j} = \begin{cases} \frac{x_i \cdot x_j}{\sqrt{d}}, & \text{if } M_{i,j} = 0, \\ -\infty, & \text{if } M_{i,j} = -\infty. \end{cases}$$

Since the embeddings are orthonormal, $x_i \cdot x_j = 0$ unless $x_i = x_j$. Therefore:

- For $i = 0$ and $j = i_h$,
$$\alpha_{0,i_h} = \frac{x_0 \cdot x_{i_h}}{\sqrt{d}}.$$

- For $i = i_h$ and $j = 0$,
$$\alpha_{i_h,0} = \frac{x_{i_h} \cdot x_0}{\sqrt{d}}.$$

- For $i = j$,
$$\alpha_{i,i} = \frac{x_i \cdot x_i}{\sqrt{d}} = \frac{\|x_i\|^2}{\sqrt{d}} = \frac{1}{\sqrt{d}}.$$

- All other
$$\alpha_{i,j} = -\infty.$$

Due to the mask and orthogonality, the attention weights after the Softmax become:

- For $i = 0$:
$$a_{0,j} = \begin{cases} 1, & \text{if } j = i_h, \\ 0, & \text{otherwise.} \end{cases}$$

- For $i = i_h$:
$$a_{i_h,j} = \begin{cases} 1, & \text{if } j = 0, \\ 0, & \text{otherwise.} \end{cases}$$

- For $i \neq 0, i_h$:
$$a_{i,j} = \begin{cases} 1, & \text{if } j = i, \\ 0, & \text{otherwise.} \end{cases}$$

Therefore, the attention score will be:

- At position $i = 0$:
$$\text{Attention}(X)_0 = a_{0,i_h} v_{i_h} = x_{i_h}.$$

- At position $i = i_h$:
$$\text{Attention}(X)_{i_h} = a_{i_h,0} v_0 = x_0.$$

- At other positions $i \neq 0, i_h$:
$$\text{Attention}(X)_i = a_{i,i} v_i = x_i.$$

$\square$

We now consider the existence of residual connection, then the final output of self-attention will be:

- At position $i = 0$:
$$u_0 = x_0 + \text{Attention}(X)_0 = x_0 + x_{i_h}.$$

- At position $i = i_h$:

$$u_{i_h} = x_{i_h} + \text{Attention}(X)_{i_h} = x_{i_h} + x_0.$$

- At other positions $i \neq 0, i_h$:

$$u_i = x_i + \text{Attention}(X)_i = x_i + x_i = 2x_i.$$

We will design the FFN such that, when considering the residual connection, the hidden state $h_i$ at each position $i$ correctly represents the updated TM configuration.

The general strategy is that, at Position $i = 0$, the FFN output $\text{FFN}(u_0)$ will be $e(q_{t+1}) - e(q_t)$, so that:
$$h_0 = u_0 + \text{FFN}(u_0) = [e(q_t) + \text{other terms}] + [e(q_{t+1}) - e(q_t)]$$
resulting in $h_0 = e(q_{t+1}) + \text{other terms}$. While at Position $i = i_h$, the FFN output $\text{FFN}(u_{i_h})$ will be $e(\gamma'_{h_t}) - e(\gamma_{h_t})$, so that:
$$h_{i_h} = u_{i_h} + \text{FFN}(u_{i_h}) = [e(\gamma_{h_t}) + \text{other terms}] + [e(\gamma'_{h_t}) - e(\gamma_{h_t})]$$
resulting in $h_{i_h} = e(\gamma'_{h_t}) + \text{other terms}$. At Other Positions $i \neq 0, i_h$, the FFN output $\text{FFN}(u_i)$ will be zero, so $h_i = u_i$, keeping the embeddings unchanged.

**Lemma A.3.** $\exists$ *a FFN s.t.* $\text{FFN}(u_0) = e(q_{t+1}) - e(q_t)$, $\text{FFN}(u_{i_h}) = e(\gamma'_{h_t}) - e(\gamma_{h_t})$, *and* $h_i = u_i$ *for* $i \neq 0, i_h$.

*Proof.* **We start with constructing the first layer.**

(1) For the neurons that is updating state embedding at $i = 0$, in which for each transition $(q_t, \gamma_{h_t}) \to q_{t+1}$, we allocate one neuron. We denote the neuron index as $n(q_t, \gamma_{h_t})$ for n $\in 1, 2, \cdots, N_{\text{trans}}$. Let the weights $w_1^{(n)}$ entries corresponding to $e(q_t)$ to be $+1$, $p(0)$ to be $+1$ and the bias $b_1^{(n)} = -1.5$, and therefore we have:
$$z_n = w_1^{(n)} \cdot u_0 + b_1^{(n)} = 1(e(q_t)) + 1(p(0)) + (-1.5) = 0.5.$$

The neuron activates (since $z_n > 0$) only if the input contains $e(q_t)$ and $p(0)$.

(2) For the neurons that is updating state embedding at $i = i_h$, in which for each transition $(q_t, \gamma_{h_t}) \to \gamma'_{h_t}$, we allocate one neuron. We denote the neuron index as $n'(q_t, \gamma_{h_t})$ in $N_{\text{trans}} + 1$ to $2N_{\text{trans}}$ Let the weights $w_1^{(n')}$ entries corresponding to $e(q_t)$ to be $+1$, $p(i_h)$ to be $+1$ and the bias $b_1^{(n')} = -1.5$, and therefore we have:
$$z_{n'} = w_1^{(n')} \cdot u_{i_h} + b_1^{(n')} = 1(e(q_t)) + 1(p(i_h)) + (-1.5) = 0.5.$$

The neuron activates only if the input contains $e(q_t)$ and $p(i_h)$.

(3) For each dimension $j$ in $d$, we allocate one neuron to pass through the input. Their neuron index $n''(j)$ are from $2N_{\text{trans}} + 1$ to $2N_{\text{trans}} + d$ The weights: $w_1^{(n'')}$ entries corresponding to $w_{1j}^{(n'')} = 1$, and $w_{1k}^{n''} = 0$ for $j \neq j$, and the bias $b_1^{(n'')} = 0$

These neurons always activates since $u_i$ has a positive component at position $j$.

**Then we construct the second layer.**

(1) For those weights mapping state update neurons to outputs, in which for neurons $n(q_t, \gamma_{h_t})$:

For those weights in $W_2$, if it is in the row corresponding to $e(q_{t+1})$, then
$$W_2^{(e(q_{t+1})), n} = 1,$$
if it is in the row corresponding to $e(q_t)$, then
$$W_2^{(e(q_t)), n} = -1.$$

(2) The same reasoning leads to: for those weights mapping tape symbol update neurons to outputs, in which for neurons $n'(q_t, \gamma_{h_t})$:

For those weights in $W_2$, if it is in the row corresponding to $e(\gamma'_{h_t})$, then
$$W_2^{(e(\gamma'_{h_t})), n'} = 1$$
if it is in the row corresponding to $e(\gamma_{h_t})$, then
$$W_2^{(e(\gamma_{h_t})), n'} = -1$$

For neurons $n''(j)$, we set $W_2^{(j),n''(j)} = 0$ (we set the output to zero for the main embeddings). We also set the other entries in $W_2$ as zero. The bias $b_2$ is always 0 in the second layer.

Therefore:

- At position $i = 0$:
$$\text{FFN}(u_0) = e(q_{t+1}) - e(q_t)$$

- At position $i = i_h$:
$$\text{FFN}(u_{i_h}) = e(\gamma'_{h_t}) - e(\gamma_{h_t})$$

- At other positions $i \neq 0, i_h$:
$$\text{FFN}(u_i) = \mathbf{0}$$

$\square$

We have eliminated the effects of residual linking here by constructing Self-Attention and FFN.

### A.2.4 ITERATIVE LAYER APPLICATION

Apply $\mathcal{T}^{(i)}$ sequentially for $S$ layers to simulate $S$ TM steps:

$$H_{t+1} = \mathcal{T}^{(1)}(H_t)$$

$$H_{t+2} = \mathcal{T}^{(2)}(H_{t+1})$$

$$\vdots$$

$$H_{t+S} = \mathcal{T}^{(S)}(H_{t+S-1})$$

Each application corresponds to one TM step, updating the Transformer's hidden state to reflect the new TM configuration.

## A.3 ENSURING ACCURATE SIMULATION

In this subsection, we address the requirements for ensuring that the transformer accurately simulates the Turing Machine (TM) without errors and overlaps. These requirements involve careful control over the encoding space, error accumulation from quantization, and the implementation of the TM's transition function. Here, we explore precision constraints on correct simulation in the case of binary representations rather than one-hot representations.

### A.3.1 UNIQUENESS OF ENCODING

To avoid overlap between the representations of different TM configurations, the hidden state dimension $d$ must be sufficiently large. Specifically, the inequality

$$d \geq |\mathbb{Q}| + k \cdot |\Gamma|,$$

ensures that the hidden state dimension $d$ is large enough to encode the current state, the tape symbols within the acceptance window, and the tape head position without collision. This guarantees that each configuration of the TM is uniquely represented in the transformer's hidden state space, minimizing the risk of two distinct TM configurations being encoded into the same hidden state. In particular, if represented in binary rather than one-hot, the lower bound of $d$ can be further compressed to

$$d \geq \log_2(|\mathbb{Q}|) + k \cdot \log_2(|\Gamma|),$$

### A.3.2 MINIMIZING QUANTIZATION ERRORS

The transformer's computations are affected by quantization errors due to finite precision. We need to ensure that these errors do not accumulate beyond an acceptable threshold $\varepsilon$. Assume that the maximum quantization error per computational step is given by:

$$\delta_q = \frac{C}{Q}$$

where $\delta_q$ is the maximum quantization error per step. $C$ is a constant dependent on the dynamic range of the variables. And $Q$ is the number of quantization levels.

Over $S_{\max}$ computational steps, the total accumulated quantization error $\epsilon_{\text{total}}$ is:

$$\epsilon_{\text{total}} = S_{\max} \cdot \delta_q = S_{\max} \cdot \frac{C}{Q}$$

To ensure that the total error does not exceed the acceptable tolerance $\varepsilon$:

$$\epsilon_{\text{total}} \leq \varepsilon \implies S_{\max} \cdot \frac{C}{Q} \leq \varepsilon$$

Solving for $Q$:

$$Q \geq \frac{C \cdot S_{\max}}{\varepsilon}$$

This inequality precisely relates $Q$, $S_{\max}$, and $\varepsilon$ without using approximate equalities.

From Lemma 4.2, we have:

$$S_{\max} = \Theta(L \cdot d \cdot k \cdot \log_2 Q).$$

Substituting $S_{\max}$ into the inequality for $Q$:

$$Q \geq \frac{C' \cdot L \cdot d \cdot k \cdot \log_2 Q}{\varepsilon}.$$

This inequality involves $Q$ on both sides. To solve for $Q$, we can consider the properties of exponential functions. Let's denote, $Q = e^x$, after simplification:

$$e^x \geq \frac{C'' \cdot L \cdot d \cdot k \cdot x}{\varepsilon},$$

where $C'' = \frac{C'}{\ln 2}$. For sufficiently large $x$, the exponential function $e^x$ dominates the polynomial term in the numerator. Therefore, to satisfy the inequality, $x$ must be large enough such that:

$$e^x \geq \text{poly}(x)$$

Since $e^x$ grows faster than any polynomial function of $x$, the inequality will hold for large $x$.

Therefore, we conclude that:

$$e^x \geq \frac{C'' \cdot L \cdot d \cdot k \cdot x}{\varepsilon}$$

implies that $x$ (and thus $Q = e^x$) must scale exponentially with $\frac{L \cdot d \cdot k}{\varepsilon}$.

Given above, we establish that:

$$Q \geq e^{\frac{C''' \cdot L \cdot d \cdot k}{\varepsilon}}$$

where $C'''$ is a constant that absorbs $C''$ and other constants.

### A.3.3 Correct Implementation of Transition Function $\delta$

For each transformer layer to simulate one TM step, it must correctly implement the TM's transition function $\delta$, which is of the form:

$$\delta(q_t, \gamma_{h_t}) = (q_{t+1}, \gamma'_{h_t}, D)$$

where $\gamma'_{h_t}$ is the new symbol to write, and $D \in \{L, R\}$ denotes the tape head movement direction.

The transformer must meet two conditions to ensure accurate simulation: (1) The Self-Attention mechanism should accurately attend to the tape symbol under the head position $h_t$ and nearby cells in the acceptance window. (2) The Feed-Forward Network (FFN) must correctly map the state and symbol $(q_t, \gamma_{h_t})$ to the new state $q_{t+1}$, updated symbol $\gamma'_{h_t}$, and head movement $D$. These requirements ensure that each transformer layer faithfully simulates the TM's transition function for every step of the simulation.

### A.4 Combining the Requirements

With the simulation construction in A.2, combining the requirements in A.3, we conclude that:

$$\forall \epsilon > 0,\ \exists \mathcal{T} \text{ with parameters } (L, d, k, Q) \text{ satisfying } \begin{cases} d \geq \log_2(|\mathbb{Q}|) + k \cdot \log_2(|\Gamma|), \\ Q \geq e^{\frac{C''' \cdot L \cdot d \cdot k}{\varepsilon}} \end{cases}$$

such that $\forall s \leq S,\ d(H_s, \phi(C_s)) \leq \epsilon$.

## B Demonstrate Lemma 4.2

Lemma 4.2 is obvious, and we will only briefly state it here. We aim to derive how the maximum number of steps $S_{\max}$ that a transformer can simulate scales with respect to the model's parameters: the number of layers $L$, the dimension $d$, the acceptance window $k$, and the quantization levels $Q$.

### B.1 Number of Layers $L$

Since each layer simulates one Turing machine (TM) step, the maximum number of steps $S_{\max}$ scales linearly with the number of layers $L$. We assume that in the transformer model simulating a TM, each layer corresponds to one computational step of the TM. At each layer, the transformer updates the TM's configuration from step $t$ to step $t + 1$. Therefore, the total number of steps that can be simulated is directly proportional to the number of layers. Therefore, the number of steps $S_{\max}$ is given by: $S_{\max} = \Theta(L)$

### B.2 Dimension $d$

Higher dimensions allow more detailed representations of the TM's configuration, reducing the risk of overlap between states. In a $d$-dimensional space, the maximum number of mutually orthogonal vectors (representing unique configurations) is at most $d$. As the simulation progresses, more distinct configurations need to be represented without interference. Therefore, the number of unique, orthogonal configurations $N_{\text{config}}$ is at most $d$. The maximum number of steps before significant overlap or interference occurs scales linearly with $d$, leading to the conclusion: $S_{\max} = \Theta(d)$

### B.3 Acceptance Window $k$

A larger acceptance window allows the transformer to process more tape symbols simultaneously. The acceptance window $k$ represents the number of positions the transformer can attend to at each

step. In simulating a TM, the tape head may need to access multiple symbols around its current position. A larger acceptance window $k$ enables the transformer to incorporate more context at each step, reducing the number of steps required to propagate information and mitigating error accumulation. The speed at which information propagates is proportional to $k$. Thus, the ability to process more symbols per step enhances the simulation's depth, giving the result: $S_{\max} = \Theta(k)$

### B.4 QUANTIZATION LEVELS $Q$

Higher quantization levels reduce numerical error, allowing more steps to be simulated before accumulated error becomes prohibitive. Quantization levels $Q$ relate to the numerical precision of the transformer's computations. The numerical precision increases logarithmically with $Q$, specifically Precision $= \log_2(Q)$. As a result, higher precision reduces the per-step numerical error $\varepsilon$, which accumulates over $S$ steps. The accumulated error after $S$ steps is approximately $S \cdot \varepsilon$. To keep the total error below a threshold $\varepsilon_{\max}$, the number of steps $S_{\max}$ is bounded by $Q$. Since the precision scales as $\log(Q)$, the maximum number of steps scales logarithmically with the quantization levels: $S_{\max} = \Theta(\log(Q))$

### B.5 SCALING RELATIONSHIP DERIVATION

From the mapping of TM configurations to hidden states and the simulation process across multiple layers, we derive the scaling relationship:

$$S_{\max} = \Theta(L \cdot d \cdot k \cdot \log(Q))$$

This relationship implies that the number of TM steps $S_{\max}$ scales linearly with the number of layers $L$, the hidden dimension $d$, and the acceptance window $k$, while it scales logarithmically with the quantization levels $Q$. More layers allow for more steps, higher dimensions enable more complex configurations, larger windows allow the processing more tape symbols, and higher precision reduces numerical errors over time.

## C PROOF OF THEOREM 4.3

*Proof.* Lemma 4.1 told us that, since state transition function $\delta$ is local, there exists a window size $k$ sufficient to capture all necessary information to perform $s$ steps of $M$. Therefore, the autoregressive Transformer, constrained by a context window size $k$, generates the output sequence $y$ in multiple refinement rounds. Each round $r$ simulates a fixed number $s$ of computational steps of $M$, updating the output from $y^{(r-1)}$ to $y^{(r)}$ by processing the relevant segment of the sequence within the window $k$. The total number of required rounds is $R = \lceil T/s \rceil$, ensuring that all computational steps are covered.

To maintain the overall approximation error within $\epsilon$, the error tolerance is distributed across the $R$ rounds, assigning an error budget $\epsilon_r = \epsilon/R$ to each round.

This ensures that the cumulative error across all rounds does not exceed $\epsilon$. We build it with the following induction: At round 0, we have $y^{(0)} = x$ correctly encodes the initial configuration $C_0$. Since no computation has been performed, the initial error is zero: $d(y^{(0)}, C_0) = 0 \leq \epsilon$. Assume that after $r - 1$ rounds, the output $y^{(r-1)}$ approximates the configuration $C_{(r-1)s}$ with error $\epsilon_{r-1} \leq (r-1)\epsilon/R$. The error introduced in round $r$ satisfies:

$$d(y^{(r)}, C_{rs}) \leq d(y^{(r-1)}, C_{(r-1)s}) + \epsilon_r \leq (r-1)\epsilon/R + \epsilon/R = r\epsilon/R.$$

Thus, the error after round $r$ is bounded by $r\epsilon/R \leq \epsilon$. The auto-regressive Transformer processes $y^{(r-1)}$ within the context window $k$ to generate $y^{(r)}$, approximating $C_{rs}$.

By induction, we establish that after each refinement round $r$, the output $y^{(r)}$ accurately represents the configuration $C_{rs}$ within the allocated error $r\epsilon/R$. Consequently, considering the termination condition after $R = \lceil T/s \rceil$ rounds, the output $y^{(R)}$ approximates the final configuration $C_T = f(x)$ with:

$$d(y^{(R)}, f(x)) \leq R \cdot (\epsilon/R) = \epsilon.$$

$\square$

## D    RADEMACHER COMPLEXITY OF K-WINDOW NEXT TOKEN PREDICTION

*Proof.* We start by expressing the empirical Rademacher complexity for $\mathcal{H}_k$:

$$\hat{\mathcal{R}}_S(\mathcal{H}_k) = \mathbb{E}_\sigma \left[ \sup_{h \in \mathcal{H}_k} \frac{1}{m} \sum_{i=1}^m \sigma_i h(x^{(i)}) \right].$$

Since each $h \in \mathcal{H}_k$ is $L_h$-Lipschitz and $\|x^{(i)}\| \le R_x \sqrt{k}$, we can bound $\hat{\mathcal{R}}_S(\mathcal{H}_k)$ using the Lipschitz property.

By Talagrand's contraction lemma(Ledoux & Talagrand, 2013), the standard results on Rademacher complexity of Lipschitz function over bounded domains, we have:

$$\hat{\mathcal{R}}_S(\mathcal{H}_k) \le \frac{L_h}{m} \mathbb{E}_\sigma \left[ \sup_{\|h\|_{\mathrm{Lip}} \le L_h} \sum_{i=1}^m \sigma_i h(x^{(i)}) \right].$$

The supremum over $h$ can be bounded using the dual norm:

$$\sup_{\|h\|_{\mathrm{Lip}} \le L_h} \sum_{i=1}^m \sigma_i h(x^{(i)}) \le L_h \left\| \sum_{i=1}^m \sigma_i x^{(i)} \right\|_*,$$

where $\| \cdot \|_*$ is the dual norm of $\| \cdot \|$. For Euclidean norms, the dual norm is also the Euclidean norm. We compute:

$$\mathbb{E}_\sigma \left\| \sum_{i=1}^m \sigma_i x^{(i)} \right\| \le \sqrt{\mathbb{E}_\sigma \left\| \sum_{i=1}^m \sigma_i x^{(i)} \right\|^2}.$$

Since $\sigma_i$ are independent Rademacher variables and $x^{(i)}$ are fixed, we have:

$$\mathbb{E}_\sigma \left\| \sum_{i=1}^m \sigma_i x^{(i)} \right\|^2 = \sum_{i=1}^m \|x^{(i)}\|^2 \le m(R_x \sqrt{k})^2 = m R_x^2 k.$$

Therefore:

$$\mathbb{E}_\sigma \left\| \sum_{i=1}^m \sigma_i x^{(i)} \right\| \le \sqrt{m R_x^2 k} = R_x \sqrt{mk}.$$

Substituting back into the Rademacher complexity expression:

$$\hat{\mathcal{R}}_S(\mathcal{H}_k) \le \frac{L_h}{m} \cdot R_x \sqrt{mk} = L_h R_x \sqrt{\frac{k}{m}}.$$

Taking the expectation over $S$:

$$\mathcal{R}_m(\mathcal{H}_k) = \mathbb{E}_S \left[ \hat{\mathcal{R}}_S(\mathcal{H}_k) \right] \le \frac{L_h R_x \sqrt{k}}{\sqrt{m}}.$$

For a neural network with depth $l_{\max}$, activation functions with Lipschitz constant $L_\phi$, and weight matrices with spectral norms bounded by $B_d$, the Lipschitz constant $L_h$ satisfies:

$$L_h \leq B_{\text{spec}} L_\phi^{l_{\max}-1}, \quad \text{where} \quad B_{\text{spec}} = \prod_{l=1}^{l_{\max}} B_l.$$

Therefore, the Rademacher complexity of $\mathcal{H}_k$ is bounded by:

$$\mathcal{R}_m(\mathcal{H}_k) \leq \frac{L_h R_x \sqrt{k}}{\sqrt{m}} = \frac{B_{\text{spec}} L_\phi^{l_{\max}-1} R_x \sqrt{k}}{\sqrt{m}}.$$

$\square$

## E  PROOF OF THEOREM 5.7

*Proof.* Using the standard generalization bound via Rademacher complexity, we have:

$$L(h) \leq \hat{L}_S(h) + 2\mathcal{R}_m(\mathcal{H}_k) + C\sqrt{\frac{\log(1/\delta)}{2m}},$$

with probability at least $1 - \delta$, where $\mathcal{R}_m(\mathcal{H}_k)$ is the Rademacher complexity of the hypothesis class $\mathcal{H}_k$.

By Lemma 5.5, we have:

$$\mathcal{R}_m(\mathcal{H}_k) \leq \frac{B_{\text{spec}} L_\phi^{l_{\max}-1} R_x \sqrt{k}}{\sqrt{m}}.$$

Substituting the Rademacher complexity into the generalization error bound:

$$L(h) \leq \hat{L}_S(h) + 2L \frac{B_{\text{spec}} L_\phi^{l_{\max}-1} R_x \sqrt{k}}{\sqrt{m}} + C\sqrt{\frac{\log(1/\delta)}{2m}}.$$

Assuming $\hat{L}_S(h) \approx 0$ (perfect empirical risk minimization), the inequality simplifies to:

$$L(h) \leq \frac{2L B_{\text{spec}} L_\phi^{l_{\max}-1} R_x \sqrt{k}}{\sqrt{m}} + C\sqrt{\frac{\log(1/\delta)}{2m}}.$$

To ensure $L(h) \leq \epsilon$, we require:

$$\frac{2L B_{\text{spec}} L_\phi^{l_{\max}-1} R_x \sqrt{k}}{\sqrt{m}} + C\sqrt{\frac{\log(1/\delta)}{2m}} \leq \epsilon.$$

Let's denote:

$$A = 2L B_{\text{spec}} L_\phi^{l_{\max}-1} R_x \sqrt{k},$$

and

$$B = C\sqrt{\frac{\log(1/\delta)}{2}}.$$

Then the inequality becomes:

$$\frac{A}{\sqrt{m}} + \frac{B}{\sqrt{m}} \leq \epsilon \quad \implies \quad \frac{A+B}{\sqrt{m}} \leq \epsilon.$$

Solving for $m$:

$$\sqrt{m} \geq \frac{A+B}{\epsilon} \quad \Longrightarrow \quad m \geq \left(\frac{A+B}{\epsilon}\right)^2.$$

Substituting back the expressions for $A$ and $B$:

$$m \geq \left(\frac{2LB_{\text{spec}}L_\phi^{l_{\max}-1}R_x\sqrt{k} + C\sqrt{\frac{\log(1/\delta)}{2}}}{\epsilon}\right)^2.$$

By simplifying:

$$m \geq \frac{1}{\epsilon^2}\left[4L^2B_{\text{spec}}^2L_\phi^{2(l_{\max}-1)}R_x^2k + 4LB_{\text{spec}}L_\phi^{l_{\max}-1}R_xC\sqrt{k}\sqrt{\frac{\log(1/\delta)}{2}} + \frac{C^2\log(1/\delta)}{2}\right].$$

$\square$

## F    PROOF OF THEOREM 5.8

*Proof.* At time step $t$, the prediction error depends on the cumulative effect of previous errors:

$$\epsilon_t \leq L_{\text{model}}\epsilon_{t-1} + \epsilon_{\text{single}},$$

where $L_{\text{model}} = B_{\text{spec}}L_\phi^{l_{\max}-1}$ is the Lipschitz constant of the model with respect to its inputs, capturing how input errors affect the output. And $\epsilon_{\text{single}}$ is the inherent error at each step due to model imperfections. We unroll the recursion to express $\epsilon_t$ in terms of $\epsilon_{\text{single}}$:

$$\epsilon_t \leq L_{\text{model}}^{t-1}\epsilon_1 + \epsilon_{\text{single}}\sum_{i=0}^{t-2}L_{\text{model}}^i.$$

Assuming $\epsilon_1 = \epsilon_{\text{single}}$ (initial error), we get:

$$\epsilon_t \leq \epsilon_{\text{single}}\left(L_{\text{model}}^{t-1} + \sum_{i=0}^{t-2}L_{\text{model}}^i\right) = \epsilon_{\text{single}}\left(L_{\text{model}}^{t-1} + \frac{L_{\text{model}}^{t-1}-1}{L_{\text{model}}-1}\right).$$

Simplifying:

$$\epsilon_t \leq \epsilon_{\text{single}} \cdot \frac{L_{\text{model}}^t - 1}{L_{\text{model}} - 1}.$$

The cumulative error is the sum over all time steps:

$$\epsilon_{\text{cumulative}} = \sum_{t=1}^{T}\epsilon_t \leq \epsilon_{\text{single}}\sum_{t=1}^{T}\frac{L_{\text{model}}^t - 1}{L_{\text{model}} - 1}.$$

$$\Downarrow$$

$$\epsilon_{\text{cumulative}} \leq \epsilon_{\text{single}} \cdot \frac{1}{L_{\text{model}} - 1}\left(\sum_{t=1}^{T}L_{\text{model}}^t - T\right),$$

where

$$\sum_{t=1}^{T} L_{\text{model}}^t = L_{\text{model}} \cdot \frac{L_{\text{model}}^T - 1}{L_{\text{model}} - 1}.$$

Therefore:

$$\epsilon_{\text{cumulative}} \leq \epsilon_{\text{single}} \cdot \frac{1}{L_{\text{model}} - 1} \left( L_{\text{model}} \cdot \frac{L_{\text{model}}^T - 1}{L_{\text{model}} - 1} - T \right).$$

This expression captures the exponential growth of errors due to the recursive dependence in the model. To ensure $\epsilon_{\text{cumulative}} \leq \epsilon$, we need:

$$\epsilon_{\text{single}} \leq \epsilon \cdot \left( \frac{1}{L_{\text{model}} - 1} \left( L_{\text{model}} \cdot \frac{L_{\text{model}}^T - 1}{L_{\text{model}} - 1} - T \right) \right)^{-1}.$$

From our Theorem 5.7, the required sample size to achieve $\epsilon_{\text{single}}$ at each time step is:

$$m \geq \frac{1}{\epsilon_{\text{single}}^2} \left[ 4L^2 B_{\text{spec}}^2 L_\phi^{2(l_{\text{max}}-1)} R_x^2 k + \text{low order term} \right],$$

Substituting the expression for $\epsilon_{\text{single}}$ Theorem5.7:

$$m \geq \left( \epsilon \cdot \left( \frac{1}{L_{\text{model}} - 1} \left( L_{\text{model}} \cdot \frac{L_{\text{model}}^T - 1}{L_{\text{model}} - 1} - T \right) \right)^{-1} \right)^{-2} \left[ 4L^2 B_{\text{spec}}^2 L_\phi^{2(l_{\text{max}}-1)} R_x^2 k + \text{low order term} \right].$$

This expression reflects the impact of error propagation on the required sample size.

We can further simplify by recognizing that when $L_{\text{model}} > 1$, $L_{\text{model}}^T$ grows exponentially, and the term involving $T$ becomes negligible in comparison. Thus, the cumulative error is dominated by:

$$\epsilon_{\text{cumulative}} \approx \epsilon_{\text{single}} \cdot \frac{L_{\text{model}}^T}{(L_{\text{model}} - 1)^2}.$$

Therefore, to keep $\epsilon_{\text{cumulative}} \leq \epsilon$, we require:

$$\epsilon_{\text{single}} \leq \epsilon \cdot \frac{(L_{\text{model}} - 1)^2}{L_{\text{model}}^T}.$$

Substituting back $L_{\text{model}} = B_{\text{spec}} L_\phi^{l_{\text{max}}-1}$, the required sample size becomes:

$$m \geq \frac{\left( B_{\text{spec}} L_\phi^{l_{\text{max}}-1} \right)^{2T}}{\epsilon^2 \left( B_{\text{spec}} L_\phi^{l_{\text{max}}-1} - 1 \right)^4} \left[ 4L^2 B_{\text{spec}}^2 L_\phi^{2(l_{\text{max}}-1)} R_x^2 k + \text{low order term} \right].$$

$\square$

## G   PROOF OF THEOREM 5.9

*Proof.* Within a single round of length $\tau = T/R$, the error at time step $t$ depends on the errors in previous steps:

$$\epsilon_t \leq L_{\text{model}}\epsilon_{t-1} + \epsilon_{\text{single}},$$

Unrolling the recursion within a round, we have:

$$\epsilon_t \leq \epsilon_{\text{single}} \sum_{i=0}^{t-1} L_{\text{model}}^i = \epsilon_{\text{single}} \cdot \frac{L_{\text{model}}^t - 1}{L_{\text{model}} - 1}.$$

The cumulative error over $\tau$ steps in a round is:

$$\epsilon_{\text{round}} = \sum_{t=1}^{\tau} \epsilon_t \leq \epsilon_{\text{single}} \sum_{t=1}^{\tau} \frac{L_{\text{model}}^t - 1}{L_{\text{model}} - 1}.$$

This sum can be simplified using geometric series, leading to:

$$\epsilon_{\text{round}} \leq \epsilon_{\text{single}} \cdot \frac{L_{\text{model}}^{\tau+1} - (\tau L_{\text{model}}) - 1 + \tau}{(L_{\text{model}} - 1)^2}.$$

The total cumulative error is the sum over all rounds:

$$\epsilon_{\text{total}} = R\epsilon_{\text{round}}.$$

To ensure $\epsilon_{\text{total}} \leq \epsilon$, we require:

$$R\epsilon_{\text{round}} \leq \epsilon.$$

Substituting the expression for $\epsilon_{\text{round}}$:

$$\epsilon_{\text{single}} \leq \epsilon \cdot \left( R \cdot \frac{L_{\text{model}}^{\tau+1} - (\tau L_{\text{model}}) - 1 + \tau}{(L_{\text{model}} - 1)^2} \right)^{-1}.$$

The required sample size to achieve $\epsilon_{\text{single}}$ is:

$$m \geq \frac{1}{\epsilon_{\text{single}}^2} \left[ 4L^2 B_{\text{spec}}^2 L_{\phi}^{2(l_{\max}-1)} R_x^2 k + \text{low order term} \right],$$

Substituting the expression for $\epsilon_{\text{single}}$:

$$m \geq \left( \epsilon \cdot \left( R \cdot \frac{L_{\text{model}}^{\tau+1} - (\tau L_{\text{model}}) - 1 + \tau}{(L_{\text{model}} - 1)^2} \right)^{-1} \right)^{-2} [A + B + C].$$

When $L_{\text{model}} > 1$ and $\tau$ is not too large, the dominant term in the numerator is $L_{\text{model}}^{\tau+1}$, and we can approximate:

$$\epsilon_{\text{round}} \approx \epsilon_{\text{single}} \cdot \frac{L_{\text{model}}^{\tau+1}}{(L_{\text{model}} - 1)^2}.$$

Thus, the total cumulative error is:

$$\epsilon_{\text{total}} \approx R\epsilon_{\text{single}} \cdot \frac{L_{\text{model}}^{\tau+1}}{(L_{\text{model}} - 1)^2}.$$

To satisfy $\epsilon_{\text{total}} \leq \epsilon$:

$$\epsilon_{\text{single}} \leq \epsilon \cdot \frac{(L_{\text{model}} - 1)^2}{R L_{\text{model}}^{\tau+1}}.$$

Substituting back into $m$:

$$m \geq \left( \epsilon \cdot \frac{(L_{\text{model}} - 1)^2}{R L_{\text{model}}^{\tau+1}} \right)^{-2} \left[ 4L^2 B_{\text{spec}}^2 L_\phi^{2(l_{\text{max}} - 1)} R_x^2 k + \text{low order term} \right]$$

$$= \left( \frac{R L_{\text{model}}^{\tau+1}}{\epsilon (L_{\text{model}} - 1)^2} \right)^2 \left[ 4L^2 B_{\text{spec}}^2 L_\phi^{2(l_{\text{max}} - 1)} R_x^2 k + \text{low order term} \right].$$

Recall that $\tau = T/R$, so:

$$L_{\text{model}}^{\tau+1} = L_{\text{model}}^{\frac{T}{R}+1}.$$

Therefore, the sample size becomes:

$$m \geq \left( \frac{L_{\text{model}}^{\frac{T}{R}+1} \cdot R}{\epsilon (L_{\text{model}} - 1)^2} \right)^2 \left[ 4L^2 B_{\text{spec}}^2 L_\phi^{2(l_{\text{max}} - 1)} R_x^2 k + \text{low order term} \right].$$

Simplifying and substituting back:

$$m \geq \frac{\left( B_{\text{spec}} L_\phi^{l_{\text{max}} - 1} \right)^{\frac{2T}{R}+2} \cdot R^2}{\epsilon^2 (B_{\text{spec}} L_\phi^{l_{\text{max}} - 1} - 1)^4} \left[ 4L^2 B_{\text{spec}}^2 L_\phi^{2(l_{\text{max}} - 1)} R_x^2 k + \text{low order term} \right].$$

$\square$

# H   PROOF OF LEMMA 6.1

*Proof.* For each $r$, with probability at least $1 - \delta_r$:

$$L^{(r)}(h^{(r)}) \leq \hat{L}_m^{(r)}(h^{(r)}) + \epsilon_r,$$

where:

$\hat{L}_m^{(r)}(h^{(r)})$ is the empirical loss on $m$ samples at round $r$, and $\epsilon_r = O\left( \frac{(B^{(r)})^2 (L^{(r)})^2}{\sqrt{m}} \right) + \sqrt{\frac{\log(1/\delta_r)}{2m}}$.

Assuming Lipschitz continuity and bounded loss functions, we can express:

$$L^{(r)}(h^{(r)}) \leq \hat{L}_m^{(r)}(h^{(r)}) + \epsilon_r + \gamma_r L^{(r-1)}(h^{(r-1)}),$$

where $\gamma_r$ quantifies the impact of errors from round $r - 1$ on round $r$.

Expand the error recursively:

$$
\begin{aligned}
L^{(r}(h^{(r)}) &\leq \hat{L}_m^{(r)}(h^{(r)}) + \epsilon_r + \gamma_r L^{(r-1)}(h^{(r-1)}) \\
&\leq \hat{L}_m^{(r)}(h^{(r)}) + \epsilon_r + \gamma_r \left( \hat{L}_m^{(r-1)}(h^{(r-1)}) + \epsilon_{r-1} + \gamma_{r-1} L^{(r-2)}(h^{(r-2)}) \right) \\
&= \hat{L}_m^{(r)}(h^{(r)}) + \epsilon_r + \gamma_r \hat{L}_m^{(r-1)}(h^{(r-1)}) + \gamma_r \epsilon_{r-1} + \gamma_r \gamma_{r-1} L^{(r-2)}(h^{(r-2)}) \\
&\quad \vdots \\
&= \sum_{k=1}^{r} \left( \left( \prod_{j=k+1}^{r} \gamma_j \right) \left( \hat{L}_m^{(k)}(h^{(k)}) + \epsilon_k \right) \right).
\end{aligned}
$$

$\square$

## I    PROOF OF THEOREM 6.2

*Proof.* The Cumulative Error:

$$
L(h) = \sum_{r=1}^{R} \lambda_r L^{(r)}(h^{(r)}) \leq \sum_{r=1}^{R} \lambda_r \sum_{k=1}^{r} \left( \left( \prod_{j=k+1}^{r} \gamma_j \right) \left( \hat{L}_m^{(k)}(h^{(k)}) + \epsilon_k \right) \right).
$$

Let $G_{r,k} = \lambda_r \prod_{j=k+1}^{r} \gamma_j$. Then:

$$
L(h) \leq \sum_{k=1}^{R} \left( \left( \hat{L}_m^{(k)}(h^{(k)}) + \epsilon_k \right) \sum_{r=k}^{R} G_{r,k} \right).
$$

Let $\Lambda_k = \sum_{r=k}^{R} G_{r,k}$. Then:

$$
L(h) \leq \sum_{k=1}^{R} \Lambda_k \left( \hat{L}_m^{(k)}(h^{(k)}) + \epsilon_k \right).
$$

## J    PROOF OF THEOREM 6.3

Consider the assumption we made that the error impact factor $\gamma = \gamma_r$ is uniform between each round, also a uniform influence factor $\lambda_r = \lambda, \forall r \in \{1, \cdots, R\}$ and a uniform lower bound $\eta \geq \hat{L}_{m,i}(h_i) + \epsilon_i$ for simplification. Denote the upper bound of cumulative error $L(h_R)$ as $\bar{L}(h_R) = \sum_{i=1}^{R} \Lambda_i \left( \hat{L}_{m,i}(h_i) + \epsilon_i \right)$.

In this case, we have:

$$
G_{r,i} = \lambda \prod_{j=i+1}^{r} \gamma_j = \lambda \gamma^{r-i}, \quad \text{since } \gamma_j = \gamma.
$$

$$
\Lambda_i = \sum_{r=i}^{R} G_{r,i} = \lambda \sum_{r=i}^{R} \gamma^{r-i} = \lambda \sum_{s=0}^{R-i} \gamma^s = \lambda \frac{1 - \gamma^{R-i+1}}{1 - \gamma}, \quad \text{for } \gamma \neq 1.
$$

From this, it can be easily obtained that, for $\gamma \neq 1$:

$$\lim_{R \to \infty} \bar{L}(h_R) = \lim_{R \to \infty} \sum_{i=1}^{R} \Lambda_i \left( \hat{L}_{m,i}(h_i) + \epsilon_i \right).$$

$$= \lim_{R \to \infty} \frac{\eta \lambda}{1 - \gamma} \sum_{i=1}^{R} \left( 1 - \gamma^{R-i+1} \right) \to \infty.$$

$\square$

## K  PROOF OF THEOREM 6.4

*Proof.* When all $\gamma_j = \gamma$, $\Lambda_i$ simplifies to:

$$\Lambda_i = \sum_{r=i}^{R} \lambda_r \gamma^{r-i}$$

This is because:

$$\prod_{j=i+1}^{r} \gamma_j = \gamma^{r-i}$$

Suppose we change $\gamma_j$ to $\gamma'$ at certain rounds $j \in H \subset \{1, 2, \ldots, R\}$.

Then we define $\Lambda_i^{\text{modified}}$:

$$\Lambda_i^{\text{modified}} = \sum_{r=i}^{R} \lambda_r \left( \prod_{j=i+1}^{r} \gamma_j \right)$$

where:

$$\gamma_j = \begin{cases} \gamma & \text{if } j \notin H \\ \gamma' & \text{if } j \in H \end{cases}$$

We can write:

$$\frac{\Lambda_i^{\text{modified}}}{\Lambda_i} = \frac{\sum_{r=i}^{R} \lambda_r \left( \prod_{j=i+1}^{r} \gamma_j \right)}{\sum_{r=i}^{R} \lambda_r \gamma^{r-i}}$$

Let's define:

$$\delta_{i,r} = \frac{\prod_{j=i+1}^{r} \gamma_j}{\gamma^{r-i}}$$

Then:

$$\frac{\Lambda_i^{\text{modified}}}{\Lambda_i} = \frac{\sum_{r=i}^{R} \lambda_r \gamma^{r-i} \delta_{i,r}}{\sum_{r=i}^{R} \lambda_r \gamma^{r-i}} = \mathbb{E}_{r \sim \mu_i}[\delta_{i,r}]$$

where $\mu_i$ is a probability distribution over $r$ defined by:

$$\mu_i(r) = \frac{\lambda_r \gamma^{r-i}}{\Lambda_i}$$

Since $\gamma_j \in \{\gamma, \gamma'\}$, we have:

$$\delta_{i,r} = \prod_{j=i+1}^{r} \frac{\gamma_j}{\gamma} = \left(\frac{\gamma'}{\gamma}\right)^{h_{i,r}}$$

where $h_{i,r}$ is the number of rounds between $i+1$ and $r$ where $\gamma_j = \gamma'$.

Thus:

$$\Lambda_i^{\text{modified}} = \Lambda_i \times \mathbb{E}_{r\sim\mu_i}\left[\left(\frac{\gamma'}{\gamma}\right)^{h_{i,r}}\right]$$

This shows that ,$\Lambda_i^{\text{modified}} \leq \Lambda_i$, since $0 \leq \frac{\gamma'}{\gamma} \leq 1$.

The cumulative error with original $\gamma$ is:

$$L(h_R) \leq \sum_{i=1}^{R} \Lambda_i \left(\hat{L}_{m,i}(h_i) + \epsilon_i\right)$$

With the modified $\gamma_j$:

$$L^{\text{modified}}(h_R) \leq \sum_{i=1}^{R} \Lambda_i^{\text{modified}} \left(\hat{L}_{m,i}(h_i) + \epsilon_i\right)$$

$$\Delta L(h_R) = L(h_R) - L^{\text{modified}}(h_R) = \sum_{i=1}^{R} \left(\Lambda_i - \Lambda_i^{\text{modified}}\right) \left(\hat{L}_{m,i}(h_i) + \epsilon_i\right)$$

Let:

$$\kappa_i = \frac{\Lambda_i^{\text{modified}}}{\Lambda_i} = \mathbb{E}_{r\sim\mu_i}\left[\left(\frac{\gamma'}{\gamma}\right)^{h_{i,r}}\right]$$

Then:

$$L^{\text{modified}}(h_R) = \sum_{i=1}^{R} \kappa_i \Lambda_i \left(\hat{L}_{m,i}(h_i) + \epsilon_i\right)$$

The overall reduction is:

$$\Delta L(h_R) = \sum_{i=1}^{R} (1 - \kappa_i) \Lambda_i \left(\hat{L}_{m,i}(h_i) + \epsilon_i\right)$$

$\square$

## L  LIPSCHITZ-CONTINUITY OF CROSS-ENTROPY

To show that the multi-class cross-entropy loss function is $L$-Lipschitz with respect to its first argument and bounded by some constant $C > 0$, we'll analyze its properties step by step.

### L.1 DEFINITION OF CROSS-ENTROPY

**Definition L.1.** *For a classification problem with $K$ classes, the multi-class cross-entropy loss is defined as:*

$$\ell(\mathbf{p}, \mathbf{y}) = -\sum_{k=1}^{K} y_k \log(p_k)$$

where $\mathbf{y} = (y_1, y_2, \ldots, y_K)$ is the one-hot encoded true label vector. So $y_k \in \{0, 1\}$ and $\sum_{k=1}^{K} y_k = 1$. And $\mathbf{p} = (p_1, p_2, \ldots, p_K)$ is the predicted probability vector from the model, where each $p_k \in (0, 1)$ and $\sum_{k=1}^{K} p_k = 1$.

Because $\mathbf{y}$ is one-hot encoded, only one term in the summation is non-zero:

$$\ell(\mathbf{p}, \mathbf{y}) = -\log(p_{k^*})$$

where $k^*$ is the index of the true class.

The loss $\ell(\mathbf{p}, \mathbf{y}) = -\log(p_{k^*})$ approaches infinity as $p_{k^*}$ approaches zero. However, in practice, the predicted probabilities are never exactly zero due to numerical stability techniques (e.g., adding a small $\varepsilon > 0$ to predictions). We therefore restrict $p_k$ to a closed interval $[\varepsilon, 1 - (K-1)\varepsilon]$ to ensure all probabilities are valid and sum to one. Since $p_{k^*} \geq \varepsilon$:

$$\ell_{\max} = -\log(\varepsilon)$$

Thus, the loss is bounded:

$$\ell(\mathbf{p}, \mathbf{y}) \leq C = -\log(\varepsilon)$$

**Definition L.2.** *A function $f$ is L-Lipschitz continuous with respect to $\mathbf{p}$ if:*

$$|f(\mathbf{p}_1) - f(\mathbf{p}_2)| \leq L \|\mathbf{p}_1 - \mathbf{p}_2\|$$

*for all $\mathbf{p}_1, \mathbf{p}_2$ in the domain, and $\|\cdot\|$ denotes a norm (e.g., Euclidean norm).*

**Theorem L.3.** *The cross-entropy loss function $\ell(\mathbf{p}, \mathbf{y})$ is L-Lipschitz continuous with respect to $\mathbf{p}$ with $L = \frac{1}{\varepsilon}$.*

*Proof.* The gradient of $\ell$ with respect to $\mathbf{p}$ is:

$$\nabla_{\mathbf{p}} \ell(\mathbf{p}, \mathbf{y}) = \left( \frac{\partial \ell}{\partial p_1}, \frac{\partial \ell}{\partial p_2}, \ldots, \frac{\partial \ell}{\partial p_K} \right)$$

Since $\ell(\mathbf{p}, \mathbf{y}) = -\log(p_{k^*})$, the partial derivatives are:

$$\frac{\partial \ell}{\partial p_k} = \begin{cases} -\frac{1}{p_k} & \text{if } k = k^* \\ 0 & \text{if } k \neq k^* \end{cases}$$

Using the Euclidean norm:

$$\|\nabla_{\mathbf{p}} \ell\| = \sqrt{\sum_{k=1}^{K} \left( \frac{\partial \ell}{\partial p_k} \right)^2} = \left| -\frac{1}{p_{k^*}} \right| = \frac{1}{p_{k^*}}$$

Since $p_{k^*} \geq \varepsilon$:

$$\|\nabla_{\mathbf{p}} \ell\| \leq \frac{1}{\varepsilon}$$

For any two probability vectors $\mathbf{p}_1$ and $\mathbf{p}_2$, there exists $\xi$ between $\mathbf{p}_1$ and $\mathbf{p}_2$ such that:

$$\ell(\mathbf{p}_1, \mathbf{y}) - \ell(\mathbf{p}_2, \mathbf{y}) = \nabla_{\mathbf{p}}\ell(\xi)^T(\mathbf{p}_1 - \mathbf{p}_2)$$

Taking absolute values:

$$|\ell(\mathbf{p}_1, \mathbf{y}) - \ell(\mathbf{p}_2, \mathbf{y})| \leq \|\nabla_{\mathbf{p}}\ell(\xi)\| \, \|\mathbf{p}_1 - \mathbf{p}_2\|$$

Using the bound on the gradient norm:

$$|\ell(\mathbf{p}_1, \mathbf{y}) - \ell(\mathbf{p}_2, \mathbf{y})| \leq \frac{1}{\varepsilon} \|\mathbf{p}_1 - \mathbf{p}_2\|$$

Therefore, the Lipschitz constant is:

$$L = \frac{1}{\varepsilon}$$

$\square$

