# OpenReview forum: "Towards Understanding Multi-Round Large Language Model Reasoning: Approximability, Learnability and Generalizability"
_ICLR.cc/2025/Conference — Submitted to ICLR 2025_

### Official Review · Reviewer_1mrS · 2024-10-28

**Soundness:** 3
**Presentation:** 2
**Contribution:** 3
**Rating:** 5
**Confidence:** 1

**Summary:**

This work analyzes the approximation, learnability, and generalization properties of multi-round auto-regressive models, including chain-of-thought, self-debate, self-refinement, and so on. This work provides a theoretical insight and analysis of the multi-round auto-regressive model.

**Strengths:**

After reading the paper, the following are the strengths.
* The solved problem is important. Previously, it was not clear whether a multi-round model works, or why it really works. This paper tries to build a theoretical analysis of the multi-round model.
* The paper analyzes multi-round models in a wind-range, including approximation, learnability, and generalization properties.

**Weaknesses:**

The following is the weakness:
* Is it possible to conduct any experiments to prove the claims? Or is there any way to prove that the analysis is correct?
* The paper only contains theoretical proof so it may not be easy for the reader to understand. Therefore, is possible, that the author could provide any other methods to support the claim, such as figures, table, or others.

**Questions:**

N/A

---

### Official Review · Reviewer_vtoS · 2024-11-03

**Soundness:** 3
**Presentation:** 3
**Contribution:** 3
**Rating:** 6
**Confidence:** 1

**Summary:**

This paper provides a theoretical analysis of multi-round reasoning in auto-regressive language models. It first shows that, from an approximation perspective, Transformer models with a limited context window size can serve as universal approximators for Turing-computable functions. Then, it shows from a learnability standpoint, PAC learning can be extended to finite-size window next-token prediction and sequence generation. Finally, it shows from a generalization perspective, generation error can propagate between rounds of multi-round generation, but proper interventions can mitigate this effect to ensure generated sequences remain within certain bounds.

**Strengths:**

- This paper provides a theoretical analysis of multi-round reasoning in large language models. The analyses are useful for understanding language models' capabilities for complex reasoning
- The paper is well written.

**Weaknesses:**

- The authors briefly discuss the practical implications of the theoretical results. It would be even more interesting if they could consider including even simple results to support their claims and demonstrate the practical usage of the theoretical foundations.

**Questions:**

- The authors discuss how proper interventions can mitigate error propagation across multi-round reasoning. However, in practice, interventions may not always be positive; for example, self-refinement can sometimes be error-prone. How would the generalization dynamics change in such cases?

---

> ### Author Response · Authors · 2024-12-03
>
> Dear Reviewer,
>
> We appreciate your time and effort in providing feedback on our submission.
>
> As the author-reviewer discussion period draws to a close, we look forward to hear whether our response addressed your concerns and are happy to engage in further discussion if there are still outstanding issues with the paper.
>
> Authors

---

> > ### Comment · Reviewer_vtoS · 2024-12-03
> >
> > Thank you to the authors for their response. It helped me better understand the paper. However, this paper is not within my area of expertise, so I will defer the decision to the other reviewers.

---

### Official Review · Reviewer_BjZn · 2024-11-04

**Soundness:** 2
**Presentation:** 3
**Contribution:** 3
**Rating:** 3
**Confidence:** 3

**Summary:**

The authors present a theoretical analysis of the approximation ability, learnability, and generalizability of multi-round transformer models (e.g., transformers augmented with prompting techniques such as chain-of-thought, etc). Though the subject is interesting and the findings potentially quite consequential (deriving sample complexities for multi-round transformer training), the paper and the proofs are difficult to follow, with many missing definitions and insufficiently unexplained proof steps.

**Strengths:**

- The authors present a theoretical characterization of the learnability of transformer models with multiple rounds. They specifically show that using multi-round inference helps to make learning more efficient, at least in principle.
- The authors also characterize the generalization ability of such models.

**Weaknesses:**

- A number of critical proof steps are unclear or not sufficiently well-explained (see questions below).
- There are some missing definitions that make it difficult for readers to understand a number of key points.

**Questions:**

I list more detailed comments and questions below. In addition, there are a small number of typos and grammatical errors, especially with respect to grammatical number agreement. I include such errors below for the first two sections, but the list is not comprehensive and I encourage the authors to re-read the manuscript more thoroughly and correct all such errors.

Line 77: "and then auto-regressive generating sequence"
  Typo/grammatical error?

Line 83: Could you please specify what parameter or variable this quantity is exponential in? Is it the sequence length, model size, or something else?

Line 117: "LLM" -> "LLMs"

Line 120: "context" -> "contexts"

Line 142: "limitation" -> "limitations"

Line 161: "long sequence generation task" -> "the long sequence generation task" or "long sequence generation tasks"

Line 216: Missing space after "where:"
  What is the meaning of the notation "q_0 x #"?

Line 219: \Gamma^* Q \Gamma^* is undefined.

Proof of 4.1: This proof is difficult to follow and understand. Many variables are undefined (e.g., What is the acceptance window k? What is h_t?). The length of the tape encoded in the hidden state must be proportional to the number of steps simulated by the model, right? How is the orthogonality of each aspect of the embedding maintained in the output layer after residual connections? Wouldn't the residual connections destroy the surjectivity of the encoding (i.e., the hidden state now encodes a mixture of two Turing machine configurations)? Why is the characterization of the transformer as a Boolean circuit necessary for the proof? In section A.5.2., Q is bounded below by 2^{\epsilon*\varepsilon/(L*d*k)}. Where does this expression come from? This inequality would imply that as the error thresholds go to zero, the lower bound on Q goes to 2^0 = 1, which is nonsensically trivial. Section A.5.3. provides a set of "requirements" or conditions on the transformer's implementation of the state transition function, but does not provide a construction of such an implementation. In general, this proof needs additional details to more clearly and precisely explain its steps, to more effectively convince the readers of its correctness.

Line 797: "Combing" -> "Combining"? (line 799 too)

Proof of 4.3: How is the output of one round of transformer computation encoded as a single output token? This output token is then appended to the input for the next round of transformer computation. Then how is the corresponding Turing machine configuration recovered from the newly-appended token to proceed with the simulation of the Turing machine?

Assumption 5.3: A comment on the Lipschitz-continuity and boundedness of the cross-entropy function would be useful here, akin to Assumptions 5.1 and 5.2.

Line 895: Missing citation.

Section 5.3: The definition of "round" here is imprecise. Does a round not correspond to a single forward pass in a transformer model? How is generating N tokens in R rounds (where each round produces N/R tokens) different from generating N tokens in a single round? Would it not require a total of N forward passes in either case? Do the rounds indicate the frequency of supervising information during the intermediate steps of sequence generation? More clarity is needed.

How the learnability analysis in Section 5 builds upon or relates to the approximation ability discussed in Section 4?
Please clarify how the generalization analysis in Section 6 connects to both the approximation and learnability results.

---

> ### Author Response · Authors · 2024-12-03
>
> Dear Reviewer,
>
> We appreciate your time and effort in providing feedback on our submission.
>
> As the author-reviewer discussion period draws to a close, we look forward to hear whether our response addressed your concerns and are happy to engage in further discussion if there are still outstanding issues with the paper.
>
> Authors

---

> ### Comment · Reviewer_BjZn · 2024-12-03
> **Response to Authors**
>
> I thank the authors for their thoughtful response.
>
> **Clarity of Lemma 4.1:** The clarity of the presentation of this lemma needs further improvement. $H_s$, $\phi$, $C'''$ are undefined. The "empirical tape space" is also undefined. Though not as major, the writing could also be further improved: e.g., line 241 is missing a period. Is $\mathcal{M}$ the same as $M$?
>
> **Proof of Lemma 4.1:** The mapping $\phi$ is not, strictly speaking, surjective. For example, if $H_s$ did not consist of one-hot vectors, it would not correspond to any input Turing machine state. For a fixed $d$, there is a maximum number of tape symbols $\Gamma_t$ that can be encoded in the hidden state, right?
>
> However, the clarity of the proof of Lemma 4.1 is much improved (perhaps the additional background on the transformer self-attention mechanism is unneeded, but not unwelcome). One thing that is not completely clear is whether the entire Turing machine state is encoded in the embedding of a single token, or across the embeddings of all tokens (such as by having each tape position correspond to a token). The equation on line 785 suggests everything is concatenated into a single large vector, but the equation on line 774 suggest each token encodes one tape position and its value. But judging from the construction's use of multi-head attention, it seems that each token encodes one tape position and its value. It is also not clear whether the concatenation operator $\oplus$ refers to vector concatenation or token concatenation.
>
> Why use the attention mask to implement the computation in the MHA block? This is rather unrepresentative of real-world transformer models which rely on the projection matrices $W_Q$, $W_K$, $W_V$ (and possibly biases) to implement the computation in this layer (and they either use a causal attention mask as in decoder-only models, or none at all, as in encoder-only models). I don't think this is fatal to the proof as I think it is possible to implement the same computation using the projections, but I strongly recommend the authors re-write this construction.
>
> The output of the attention layer (after residuals) is identical at position $0$ and $i_h$ (i.e., $u_0 = u_{i_h}$), and therefore $FFN(u_0)$ must be equal to $FFN(u_{i_h})$. $\gamma_{h_t}'$ is also undefined.
>
> I don't see how the construction of one MHA layer helps to implement a single step in a Turing machine. One construction that may work is to have the attention mechanism shift the tape by one position to the left or right, depending on the state and value at the head position. The FFN would then need to implement the state transition function and perform any write operation to the tape at the head position. But it seems from the current description in the proof, the MHA layer aims to swap the states at position $0$ and $i_h$ (but the aim of the FFN layer seems to align with this suggested construction).
>
> Line 1022: "In particular, if represented in binary rather than one-hot, the lower bound of d can be further
> compressed to..."
>
> This would make the construction in Section A.2 invalid, and you would need to define a new construction to work with the non-one-hot encodings.
>
> How does the construction here relate to that of [1]? They similarly show that transformers with chain-of-thought can simulate Turing machines.
>
> [1] William Merrill and Ashish Sabharwal. The expressive power of transformers with chain of thought. ICLR 2024.
>
> Overall I think the premise of the work is interesting and the portions on learnability are potentially very impactful. However, I do think the paper needs an additional round of revision to further tighten the arguments and the writing. As such, I will maintain my overall score, but I will slightly increase the presentation score in recognition of the improved clarity so far.

---

### Official Review · Reviewer_M6Wa · 2024-11-04

**Soundness:** 2
**Presentation:** 4
**Contribution:** 2
**Rating:** 5
**Confidence:** 4

**Summary:**

In the paper, the authors aim to show that Transformers are universal approximators and that multi-round generation is learnable.

**Strengths:**

Well written paper.

**Weaknesses:**

In the current version, I do not see a significant contribution to ICLR.

I see the key contributions, Lemma 4.1 and Theorem 4.3, and the following basic assumptions and Lemmas, as not significant and novel for the following reasons:
1) The paper overall and section 4 in particular vastly ignores previous work, i.e. the contributon by Hava Siegelman in the 1990's showing (and proofing) that RNNs are super-Turing for two main arguments: continuous weight space and continuous activation functions. A transformer is an extension of an RNN (with a finite time window) with generalised access to information due to the attention mechanism. Thus, it is obvious that a Transformer exists that can simulate an arbitrary TM.
2) The paper provides a trivial sketch for Lemma 4.1 but not a sound mathematical proof.
3) Related to 1), the theoretical computational complexity of RNNs and Transformers is long known, as an RNN/Transformer can solve any problem in NP, but thus it is impossible to find the respective Transformer configuration. In fact, the main difficulty is to show that the available training and in-context learning or prompting mechanisms can yield such configurations. However, this is not sufficiently covered in the paper (i.e. section 5), as the focus lies on the complexity of the information/sample, but not on the currently used training algorithm.

**Questions:**

none.

---

> ### Author Response · Authors · 2024-12-03
>
> Dear Reviewer,
>
> We appreciate your time and effort in providing feedback on our submission.
>
> As the author-reviewer discussion period draws to a close, we look forward to hear whether our response addressed your concerns and are happy to engage in further discussion if there are still outstanding issues with the paper.
>
> Authors

---

### Meta-Review · Area_Chair_44aT · 2024-12-17

**Metareview:**

This paper presents a theoretical analysis of multi-round reasoning methods like chain-of-thought prompting. It is outside of my expertise and I defer to the reviewers’ reviews, see below.

The reviewers found the paper to be potentially significant (BjZn, vtoS, 1mrS). It is not within the area of expertise of 1mrS and vtoS who would welcome simple experiments to illustrate the claims. BjZn and the authors discussed the analysis in detail but overall, BjZn felt that the paper needed an additional round of revision to tighten the arguments and writing. M6Wa had concerns about the novelty of the paper which the authors addressed in their rebuttal.

**Additional Comments On Reviewer Discussion:**

-

---

### Decision · Program_Chairs · 2025-01-22

Reject